



# Aero-elastic load validation in wake conditions using nacelle-mounted lidar measurements

Davide Conti[1], Nikolay Dimitrov[1], and Alfredo Peña[1]

[1]Department of Wind Energy, Technical University of Denmark, Frederiksborgvej 399, 4000 Roskilde, Denmark

**Correspondence:** Davide Conti (davcon@dtu.dk)

**Abstract.** We propose a method for carrying out wind turbine load validation in wake conditions using measurements from forward-looking nacelle lidars. Two lidars, a pulsed and a continuous wave system, were installed on the nacelle of a 2.3 MW wind turbine operating in free-, partial- and full-wake conditions. The turbine is placed within a straight row of turbines with a spacing of 5.2 rotor diameters and wake disturbances are present for two opposite wind direction sectors. We account for wake-induced effects by means of wind field parameters commonly used as inputs for load simulations, which are reconstructed using lidar measurements. These include mean wind speed, turbulence intensity, vertical and horizontal shear, yaw error and turbulence-spectra parameters. The uncertainty and bias of aero-elastic load predictions are quantified against wind turbine on-board sensor data. We consider mast-based load assessments in free wind as a reference case and assess the uncertainty in lidar-based power and load predictions when the turbine is operating in partial- and full-wake. Compared to the reference case, the simulations in wake conditions lead to an increase of the relative error as low as 4%. It is demonstrated that the mean wind speed, turbulence intensity and turbulence length scale have a significant impact on the predictions. Finally, the experiences from this study indicate that characterizing turbulence inside the wake as well as defining a rotor equivalent wind speed model are the most challenging aspects of load validation in wake conditions.

## 1 Introduction

Wind turbines are designed according to reference wind conditions described in the IEC standards (IEC, 2019). These reference conditions are used to establish the full design load basis and for the purpose of certification of turbine designs. Nevertheless, certified turbines need to be further verified that they can withstand the site-specific loads during the entire lifetime, when site-conditions exceed those of the type-certified. As a current best practice, the wind turbine (WT) operating loads are predicted using high-fidelity aero-elastic simulations based on site-specific environmental conditions. The environmental conditions are typically obtained from anemometers installed on meteorological masts in the proximity of the wind turbine location. These mast measurements, and therefore the uncertainty quantification of the aero-elastic model, are usually limited to wake free sectors. However, wind conditions inside wind farms are significantly different than those in undisturbed wind conditions





(Frandsen, 2007). Wake effects are responsible for wind speed reduction and turbulence levels increase, generally resulting in reduced power productions and increased load levels (Larsen et al., 2013). To account for these effects, aero-elastic load simulations are combined with wake models, which predict wake-induced effects on the flow field approaching individual WTs. The most applied approach consists of increasing the turbulence in load simulations, resulting in a load increase which

should correspond to the effect of the wake-added turbulence. The effective turbulence depends on the park layout and on the material properties of the turbine components under consideration (Frandsen, 2007). This approach is recommended by the IEC 61400-1. An alternative and more detailed practice also described in the IEC standard relies on the use of the Dynamic Wake Meandering model (DWM) (Larsen et al., 2006, 2007; Madsen et al., 2010), which is an engineering model providing simulated wind field time series including wake deficits. To date, these approaches are characterized by a significant level of uncertainty,

due to the stochastic nature of environmental conditions and the various simplifying assumptions used in the wake model definitions (Schmidt et al., 2011; Reinwardt et al., 2018). The recent applications of lidars in the wind energy field demonstrate the feasibility of these systems to reconstruct inflow wind conditions including mean wind speed (Raach et al., 2014; Borraccino et al., 2017), turbulence (Mann et al., 2009; Branlard et al., 2013; Peña et al., 2017) and wake characteristics (Bingöl et al., 2010; Machefaux et al., 2016), among others. Nacelle-mounted lidars enable us to measure wind field characteristics for any

wind direction/nacelle yaw position, including situations when the turbine rotor is in the wake of a neighbouring turbine. An excellent level of agreement has been found between the nacelle-mounted lidar-estimated and mast-measured mean wind speed in free wind conditions (Borraccino et al., 2017). Power curve validations using nacelle-mounted lidars have been showing promising results (Wagner et al., 2014). Although lidar-derived along-wind variances could deviate from those derived from cup anemometer measurements (Peña et al., 2017), the load predictions in wake-free sectors based on nacelle-lidar wind field

representations resulted in uncertainties lower than or equal to those obtained with mast measurements (Dimitrov et al., 2019). Based on these findings, we extend the load validation procedure defined in Dimitrov et al. (2019) to include wake conditions. The objective of this study is to demonstrate how loads in wake conditions can be predicted accurately, quantify the uncertainty, and compare it to the uncertainty of mast-based load assessments in free wind. The further development of lidar-based load and power validation procedures can potentially replace the use of expensive meteorological masts in measurement campaigns

as well as improve the wake field reconstruction for aero-elastic load simulations.

The paper is structured as follows. In Sect. 2, we introduce the requirements for load validation and describe the measurement campaign. In Sect. 3, we present the methods implemented to derive the wind field parameters for aero-elastic simulations and a wake detection algorithm. The results are provided in Sect. 4. First, we show the wake-induced effects on the lidar-estimated wind field parameters in Sect. 4.1. Then, we derive the wind field characteristics used as input for load simulations in Sect.

4.2. The uncertainties of load predictions are quantified in Sect. 4.3 and a sensitivity analysis to assess the influence of inflow parameters on load predictions is carried out in Sect. 4.4. Finally, we discuss the findings and provide conclusions in the last two sections.





## 2 Problem formulation

### 2.1 Requirements for load validation in wakes

The design load cases and load validation procedure for wind turbines are described in the IEC standards. The IEC61400-1 requires the evaluation of fatigue and extreme loading conditions induced by wake effects originating from neighboring wind turbines. The increase in loading due to wake effects can be accounted for by the use of an added turbulence model, or by using more detailed wake models (i.e., DWM). Load validation guidelines are described in IEC61400-13 (IEC, 2015), which recommends the so-called one-to-one comparison, among few approaches. This approach consists of carrying out individual aeroelastic simulations for each measured realization of environmental conditions. To date, wind conditions are obtained from meteorological masts. The objective of this work is to carry out load validation of wind turbines operating in wake conditions using measurements from nacelle-mounted lidars only. The wake-induced effects are accounted for by lidar-estimated wind field characteristics, without employing wake deficit models. This implies that wake flow fields can be described by means of average flow characteristics commonly used as inputs for load simulations. We assess the viability of the suggested approach by carrying out a load validation study as following:

- One-to-one load comparison between measured and predicted load realizations using wind field characteristics derived from lidar measurements of the wake flow field.

- Uncertainty quantification in terms of the statistical properties of the ratios between measured and predicted load realizations.

- Comparison of lidar-based load predictions uncertainties in wakes against uncertainties of load predictions in free wind conditions using mast measurements.

We assume that the observed deviations in load predictions between those that are lidar-based under wake conditions and those that are mast-based under free wind conditions are solely due to the error in the wind field representation. This is a simplistic but conservative assumption, as the uncertainties of load predictions are a combination of uncertainty in the reconstructed wind profiles, aero-elastic model uncertainty, load measurement uncertainty as well as statistical uncertainty (Dimitrov et al., 2019).

### 2.2 Measuring campaign

Wind and load measurements are collected from an experiment conducted at the Nørrekær Enge (NKE) wind farm during a period of 7 months between 2015 and 2016. The farm is located in the North-west of Denmark and consists of 13 Siemens 2.3 MW turbines, with a 93-m rotor diameter (D) and hub height of 80 m above ground level. The turbines are installed in a single row oriented along the 75° and 255° direction compared to true north, with 487-m (5.2 D) spacing, as pictured in Fig. 1. The wind farm is located over flat terrain and the surface is characterized by a mix between croplands and grasslands, and a fjord to the north (Peña et al., 2017). The prevailing wind direction is west (Borraccino et al., 2017). The wind turbine T04 was



instrumented with sensors for load measurements at the roots of two blades, tower top and tower bottom (Vignaroli and Kock, 2016). Additional data were provided by the supervisory control and data acquisition (SCADA) system including nacelle wind speed and orientation, power output, blade pitch angles, and generator speed. A meteorological mast was installed at 232 m (2.5 D) distance from T04 in direction 103°. The mast instrumentation comprises cup and sonic anemometers, wind vanes

and thermometers mounted at several heights, among others. Details about the instrumentations can be found in Vignaroli and Kock (2016) and Borraccino et al. (2017). This study uses wind measurements from the cup anemometers at 57.5 and 80 m, which are used to derive wind speed, turbulence and shear as discussed in the following sections. According to the definition in IEC61400-12-1 (IEC, 2017), the wake-free sector spans approximately 123°, from 97° to 220°. A narrow sector of 12° span from 97° to 109° is chosen as free wind reference to ensure close correspondence between lidar- and mast-measured

parameters. Based on the farm geometry and visual inspection of data, wake sectors of 30° are considered ranging from 55° to 85° for north-east directions and from 235° to 265° for south-west.

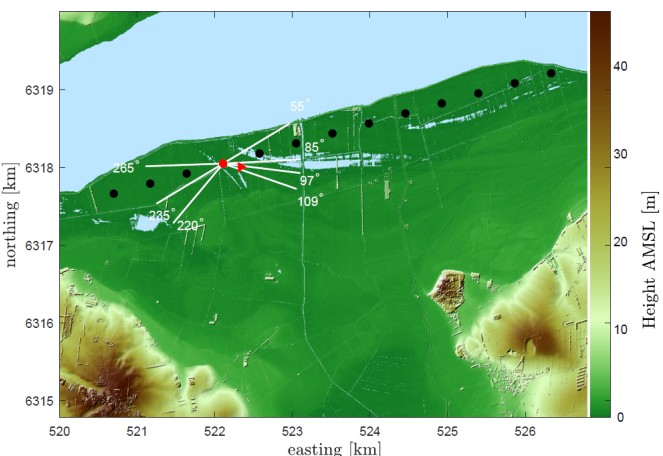

**Figure 1.** The Nørrekær Enge wind farm in northern Denmark on a digital surface elevation model (UTM32 WGS84). The wind turbines are shown in circles, the turbine T04 with the nacelle lidars in red and the mast in a triangle. The sectors used for the analysis are also shown; narrow direction sector: 97° – 109°; wide direction sector: 97° – 220°; wake sectors: 55° – 85° and 235° – 265°. The waters of Limfjorden are shown in light blue.

## 2.3 Lidars

Two forward-looking lidars were installed on the nacelle of T04: a pulsed lidar (PL) with a 5-beam configuration and a continuous wave (CW) system. The CW lidar by ZephIR has a single beam, which scans conically with a cone angle of 15°

and a sampling frequency of 48.8 Hz. The CW lidar measured sequentially at five different ranges upwind from the turbine, at 0.1, 0.3, 1.0, 1.3 and 2.5 D, and took approximately 50 s to complete a full scan at all ranges. The CW lidar measurements are binned according to the azimuthal positions in 50 bins of 7.2°. Based on Dimitrov et al. (2019), we select 10 of these bins for further analysis and focus on ranges between 0.3 and 2.5 D, as illustrated in Fig. 2 (bottom-left). The PL lidar provided





by Avent technology has five fixed beams; a central beam oriented in the longitudinal direction at hub height and four beams oriented at the corner of a square pattern, as shown in Fig. 2 (bottom-right). The PL lidar measures simultaneously at ten different ranges in front of the turbine 0.53, 0.77, 1.03, 1.17, 1.30, 1.53, 1.78, 2.03, 2.5, and 3.0 D, by acquiring radial velocity spectra for 1 s at each beam, thus scanning a single plane with a sampling frequency of 0.2 Hz (Peña et al., 2017). To provide a

5   direct comparison with results from the CW lidar, we focus the analysis on the PL lidar measurements up to 2.5 D. More details of the lidars are described in Peña et al. (2017) and Dimitrov et al. (2019), while calibration reports are provided in Borraccino and Courtney (2016a, b). The top views of the PL scanning pattern and CW lidar binned data selection are illustrated in Fig. 2 (top). The lidars measure approximately within 2.5 and 5 D downstream of the wake-source turbine. We conduct the load analysis using 10-min reference periods. The dataset is filtered so that we select only periods where the turbine is operational

10   and load, mast and lidar measurements are available. A total of 6198 10-min periods are available in the wide direction sector, which reduces to 1042 samples in the narrow sector. The majority of measurements within the wake sectors are from westerly directions $235° - 265°$ with 3659 samples, while 899 samples are available from wake directions $55° - 85°$.

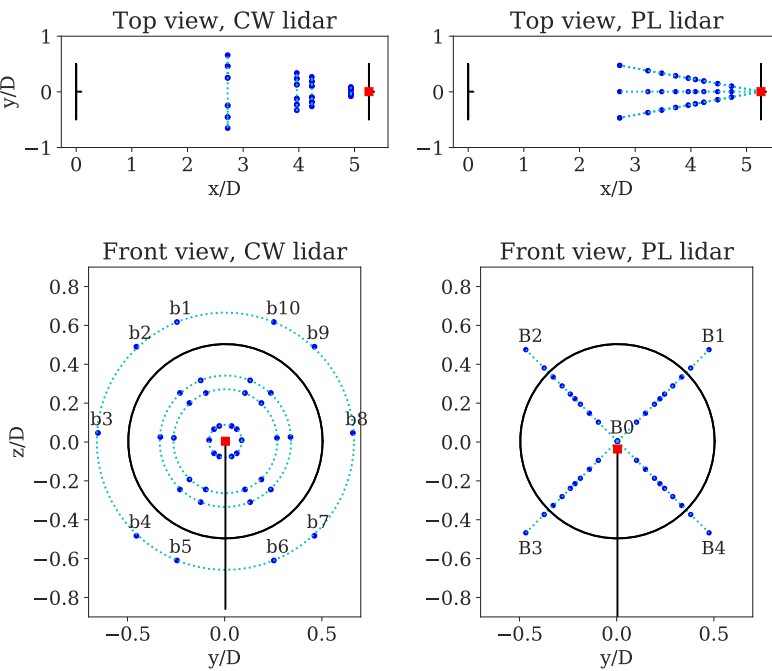

**Figure 2.** Top and front views of the CW lidar (left) and PL lidar (right) scanning patterns shown by the blue dots. The trajectory of the lidars beams is illustrated by the dotted lines in cyan. The bins/beams notation is also given. The location of the lidars on T04 is shown with a red square marker. The reference coordinate system has an origin at the hub center with the *x*-axis is in the mean wind direction. The distances are normalized with respect to the rotor diameter D.





## 3 Methodology

Load simulations are carried out using the state-of-the-art aero-elastic HAWC2 software (Larsen and Hansen, 2007). The turbine model is based on the structural and aerodynamic data of the Siemens SWT 2.3-93 turbine and is equipped with the original equipment manufacturer controller. The turbulence used in the simulations is generated using the Mann turbulence model (Mann, 1994). As described in Dimitrov et al. (2018), the turbulent wind field for aero-elastic simulations can be fully characterized statistically by nine environmental parameters listed in Table 1. The methods to derive the wind field parameters from the radial velocity measurements of the nacelle-mounted lidars are described in Sects. 3.1–3.3. We propose a wake detection algorithm to detect wakes using lidar measurements in Sect. 3.4.

**Table 1.** Wind field parameters serving as input for aeroelastic load simulations.

| Description | Parameter | Description | Parameter |
|---|---|---|---|
| Mean wind speed at hub height | $\bar{u}_{hub}$ | Air density | $\rho$ |
| Turbulence intensity | $\sigma_U/U$ | Mann turbulence spectra tensor parameters: | |
| Shear exponent | $\alpha$ | turbulence length scale | $L$ |
| Wind veer | $\Delta\varphi$ | anisotropy factor | $\Gamma$ |
| Yaw misalignment | $\bar{\varphi}$ | turbulence dissipation parameter | $\alpha_k\epsilon^{2/3}$ |

### 3.1 Wind field reconstruction

Wind field reconstruction (WFR) is defined as the process of retrieving wind field characteristics by combining measurements of the wind in multiple locations (Raach et al., 2014; Borraccino et al., 2017). As nacelle-mounted lidars measure only the line-of-sight (LOS) component of the wind vector, WFR techniques are used to derive the input wind field variables for carrying out load simulations. The present work implements the WFR technique described in Dimitrov et al. (2019). This approach assumes three-dimensional wind vectors, vertical and horizontal wind profiles combined with an induction model. The vertical wind shear is defined by a power law profile,

$$\bar{u}(z) = \bar{u}_{hub}\left(\frac{z}{z_{hub}}\right)^{\alpha}, \tag{1}$$

where $z_{hub}$ is the hub height. The flow direction $\varphi(z)$ is described by the combined effects of the mean yaw misalignment and the change of wind direction with height, the wind veer,

$$\varphi(z) = \bar{\varphi} + \frac{\Delta\varphi}{D}(z - z_{hub}). \tag{2}$$

We assume a linear variation of wind direction over the rotor diameter D. To define the relation between the free-flow wind vector $\boldsymbol{u} = (u, v, w)$ and the LOS velocity $u_{LOS}$, we consider a reference coordinate system with origin at hub height and co-linear with the wind turbine orientation. The wind coordinate system is aligned with the mean wind direction, which is defined





by the flow direction in Eq. (2). Thus, the transformation from the wind- into the reference-coordinate system is achieved by the rotational transformation $\boldsymbol{T_1}$:

$$
\boldsymbol{T_1} = \begin{bmatrix} \cos\varphi(z) & -\sin\varphi(z) & 0 \\ \sin\varphi(z) & \cos\varphi(z) & 0 \\ 0 & 0 & 1 \end{bmatrix}.
\tag{3}
$$

Note that the wind flow inclination (tilt) is neglected. The orientation of the LOS velocity with respect to the reference
coordinate system is defined by rotations about the $y$- and $z$-axes, $\psi_y$ and $\psi_z$ . Therefore, the transformation from the LOS-
into the reference-coordinate system is achieved by the rotational transformation $\boldsymbol{T_{LOS}}$:

$$
\boldsymbol{T_{LOS}} = \begin{bmatrix} \cos\psi_y \cos\psi_z & -\cos\psi_y \sin\psi_z & \sin\psi_y \\ \sin\psi_z & \cos\psi_z & 0 \\ -\sin\psi_y \cos\psi_z & \sin\psi_y \sin\psi_z & \cos\psi_y \end{bmatrix}.
\tag{4}
$$

Eventually, the relation between the wind vector and the LOS velocity is expressed in terms of matrix transformations as

$$
\boldsymbol{u_{los}} = \boldsymbol{T_{LOS}} \boldsymbol{T_1} \boldsymbol{u}.
\tag{5}
$$

This formulation is suitable assuming lidar point-like measurements and homogeneous wind field, which implies that the
three velocity component statistics do not change over the scanned area. As lidars measure only the LOS velocity component,
the first row alone of $\boldsymbol{T_{LOS}}$ is considered. By combining Eqs. (1)-(5) and including an induction factor $C_{ind}$ based on a two-
dimensional induction model (Dimitrov et al., 2019), the relation between the LOS and the wind velocity field is derived in its
extended form as:

$u_{LOS} =$

$$
= \bar{u}_{hub} \left( \frac{z_1}{z_{hub}} \right)^\alpha C_{ind} \left[ \cos\left( \bar\varphi + \frac{\Delta\varphi}{D}(z - z_{hub}) \right) \cos\psi_y \cos\psi_z - \sin\left( \bar\varphi + \frac{\Delta\varphi}{D}(z - z_{hub}) \right) \cos\psi_y \sin\psi_z \right].
\tag{6}
$$

The parameters $(\bar{u}_{hub}, \alpha, \Delta\varphi, \bar\varphi)$ are to be characterized by the WFR, while $x, y, z$ describe the spatial location of the mea-
surement points. The WFR approach relies on a model-fitting technique and consists in minimizing the residual between the
modelled wind field and lidar measurements (Borraccino et al., 2017). The CW and PL lidar-estimated mean wind speed in free
wind, for the narrow direction sector ($97°$– $109°$), are compared with measurements from the 80 m cup anemometer mounted
on the mast in Fig. 3 (left and middle). An excellent agreement is found for the lidar-estimated mean wind speed using both
lidars. The lidar-estimated shear exponents are compared with the shear obtained by fitting the power law profile using mea-
surements from the cups at 57.5 m and 80 m in Fig. 3 (right). The observed deviations result from the use of different parts of
the rotor span by the PL lidar compared to the mast measurements (Dimitrov et al., 2019). Besides, the shear exponents derived
by the CW lidar compare very well with those from the PL lidar (not shown).

## 3.2  Turbulence spectral model

The wind field vector $\boldsymbol{u}(\boldsymbol{x})$ can be described by the solely spatial vector $\boldsymbol{x} = (x, y, z)$, assuming Taylor's frozen turbulence
hypothesis (Mizuno and Panofsky, 1975). The velocity fluctuations, denoted by $\boldsymbol{u} = (u, v, w)$, are expected homogeneous in





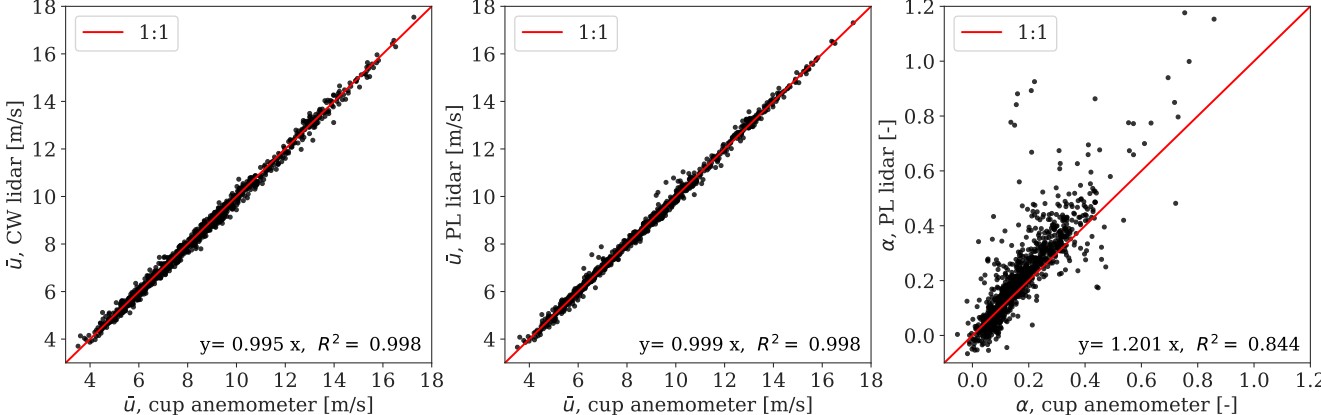

**Figure 3.** Comparison of 10-min lidar-estimated and mast-measured inflow characteristics. We show a 1:1 line for guidance, the slope of a linear regression model and the coefficient of determination $R^2$.

space (Mann, 1994). The covariance tensor of single-point turbulent statistics can be written as:

$$
\boldsymbol{R} = \begin{bmatrix} \sigma_u^2 & \sigma_{uv} & \sigma_{uw} \\ \sigma_{vu} & \sigma_v^2 & \sigma_{vw} \\ \sigma_{wu} & \sigma_{wv} & \sigma_w^2 \end{bmatrix},
\tag{7}
$$

where the matrix elements define variances and covariances of the three-dimensional velocity field $\boldsymbol{u} = (u, v, w)$. The spectral velocity tensor $\Phi_{ij}(\boldsymbol{k})$ is defined as the Fourier transform of the covariance tensor,

$$
\Phi_{ij}(\boldsymbol{k}) = \frac{1}{(2\pi)^3} \int R_{ij}(\boldsymbol{k}) \exp(i\boldsymbol{k} \cdot \boldsymbol{r}) d\boldsymbol{r},
\tag{8}
$$

where $\boldsymbol{r} = (r_1, r_2, r_3)$ is the separation vector in the three-dimensional Cartesian coordinate system and $\boldsymbol{k} = (k_1, k_2, k_3)$ is the wave number vector. The spectral velocity tensor can be described by the model of Mann (1994). This model requires only three parameters: $\alpha_k \epsilon^{2/3}, L$ and $\Gamma$, where $\alpha_k$ is the spectral Kolmogorov constant, $\epsilon$ is the turbulent energy dissipation rate, $L$ is a length scale proportional to the size of turbulence eddies, and $\Gamma$ is a parameter describing the anisotropy of the turbulence.

Although the Mann model assumes near-neutral atmospheric conditions, the model has been applied to different surface and atmospheric-stability conditions (Peña et al., 2010). The one-point spectra are computed as:

$$
F_{ij}(k_1) = \int\int \Phi_{ij}(\boldsymbol{k}, \Gamma, L, \alpha_k \epsilon^{2/3}) dk_2 dk_3.
\tag{9}
$$

The procedure to derive spectral parameters from the measured spectra of the three velocity components is described in Mann (1994). The LOS spectra measured by a lidar beam can be related to the velocity spectral tensor by accounting for probe

volume effects as described in Mann et al. (2009),

$$
F_{LOS}(k_1) = n_i n_j \int\int \mid \hat{\phi}(\boldsymbol{k} \cdot \boldsymbol{n}) \mid^2 \Phi_{ij}(\boldsymbol{k}, \Gamma, L, \alpha_k \epsilon^{2/3}) dk_2 dk_3,
\tag{10}
$$



where $\hat{\phi}$ is the Fourier transform of the lidar spatial weighting function and $\boldsymbol{n}$ is the unity vector along the beam. For a CW lidar, this is typically described by a Lorentzian function (Sonneschein and Horrigan, 1971; Mann et al., 2010). For the pulsed lidar, we assume a Gaussian weighting function (Frehlich, 2013).

### 3.3 Turbulence characterization

Turbulence characterization using lidars is subjected to several sources of uncertainty. The measurement volumes along the LOS lead to spatial averaging of turbulence, which reduces the LOS variance when compared to a point measurement (Sjöholm et al., 2008; Sathe and Mann, 2013). Besides, the ability to properly measure the variances of the velocity components depends on the scanning strategy. Since the lidar beams are rarely aligned with any of the three velocity components, the LOS variance can be influenced by the variance of other velocity components, also referred to as cross-contamination effects. We implement

two approaches to derive filtered and unfiltered turbulence based on the work of Peña et al. (2017). The first approach uses the turbulence spectral model by Mann to correct turbulence estimates by accounting for the expected attenuation of the fluctuations of the radial velocity due to the lidar's probe volume. This can be achieved numerically by deriving the relation between the variance of the LOS velocity with and without filtering effects, respectively $\sigma^2_{u,LOS,va}$ and $\sigma^2_{u,LOS,pt}$. The filtering is expressed by:

$$r(Z_r, L, \Gamma, \psi_y, \psi_z)^2 = \frac{\int_0^\infty F_{LOS}(k_1)dk_1}{\int_0^\infty F_{ij}(k_1)dk_1} = \frac{\sigma^2_{u,LOS,va}}{\sigma^2_{u,LOS,pt}}. \tag{11}$$

The magnitude of $r^2$ varies in relation to the probe volume length $Z_r$, turbulence characteristics and spatial location of measurement points (Mann et al., 2010; Peña et al., 2017). Following the procedure described in Dimitrov et al. (2019), the covariance matrix of the filtered LOS velocity components $\boldsymbol{R_{LOS}}$ can be related to the covariance of the undisturbed wind field $\boldsymbol{R}$. To express the LOS variance as function of the $u$-component variance, we normalize $\boldsymbol{R}$ with respect to $\sigma^2_U$. We neglect

the terms $\sigma^2_{uv}$ and $\sigma^2_{vw}$ as they are small and we lack sufficient information to recover all components. Hence, we derive the ratios between variances of different velocity components using the spectral tensor model by Mann (1994),

$$\frac{\boldsymbol{R}}{\sigma^2_u} = \begin{bmatrix} 1 & 0 & \sigma_{uw}/\sigma^2_u \\ 0 & \sigma^2_v/\sigma^2_u & 0 \\ \sigma_{wu}/\sigma^2_u & 0 & \sigma^2_w/\sigma^2_u \end{bmatrix}. \tag{12}$$

The effects of cross-contamination and flow direction are accounted for by means of matrix transformations including $\boldsymbol{T_{LOS}}$ and $\boldsymbol{T_1}$. The relation between the covariance matrix of the LOS is then expressed in terms of $\sigma^2_U$ as:

$$\frac{\boldsymbol{R_{LOS}}}{\sigma^2_u} = r(Z_r, L, \Gamma, \psi_y, \psi_z)^2 \left( \boldsymbol{T_{LOS}} \boldsymbol{C} \boldsymbol{T_1} \frac{\boldsymbol{R}}{\sigma^2_u} \boldsymbol{T_1^T} \boldsymbol{C^T} \boldsymbol{T_{LOS}^T} \right), \tag{13}$$

where $\boldsymbol{C}$ is the induction matrix (Dimitrov et al., 2019). Since only LOS velocities are measured by the nacelle-mounted lidar, the ratio in Eq. (13) identifies the relation between the LOS variance and the wind field variance in the longitudinal direction. Eventually, the variance of the wind field is computed by scaling the variance of the LOS residuals with the reciprocal of the filtering ratio estimated using Eq. (13). The procedure is described in details in Dimitrov et al. (2019). The second





approach avoids filtering effects by use of the ensemble-averaged Doppler radial velocity spectrum (Mann et al., 2010). This method relies on the hypothesis that the lidar average Doppler spectrum is related to the probability density function of the radial velocities (Branlard et al., 2013). This assumption is valid for homogeneous flow and for negligible velocity gradients within the probe volume. By assuming homogeneous wind flow, we use the scanning pattern to account for cross-contamination

of different velocity components and we extract 10-min $\sigma_u^2$ statistics by computing the variance of Eq. (5). We refer the reader to Eq. (10) in Peña et al. (2017) for the extended expression.

We show the comparison between lidar-estimated and mast-measured $\sigma_u$, using the 80-m cup anemometer, for free wind and narrow direction sector ($97° - 109°$) in Fig. 4. The filtered turbulence derived from CW and PL lidars are plotted respectively in Fig. 4 (left and middle), whereas the unfiltered turbulence derived from the CW lidar are shown in Fig. 4 (right). The deviations

between PL lidar and the cup anemometer values are mostly due to high-frequency noise contamination as described in Peña et al. (2017). Considering the wind conditions within the free wind narrow direction sector, the lidar-estimated turbulence compares very well with mast measurements; the observed results are consistent with previous findings (Dimitrov et al., 2019).

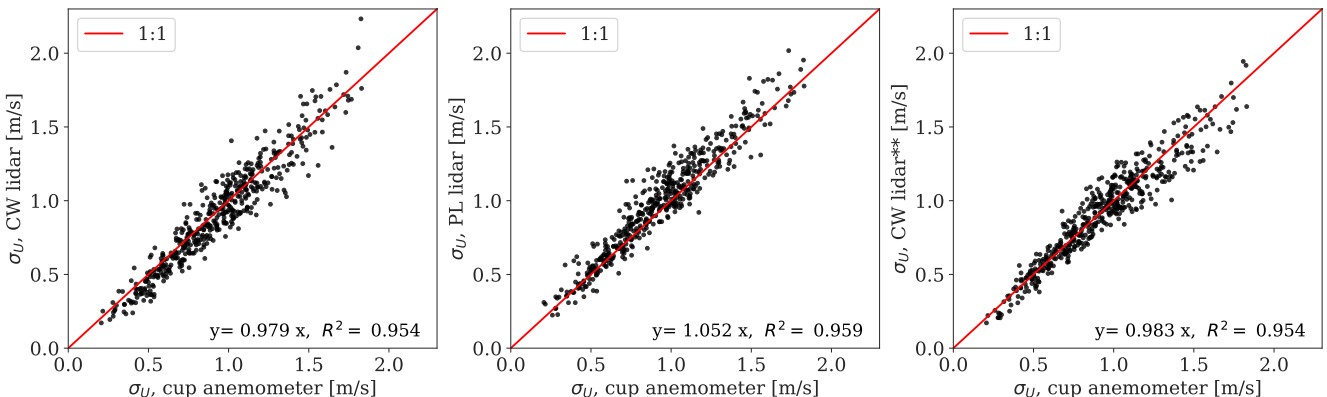

**Figure 4.** Left and middle: comparison of the lidar-based $\sigma_U$, derived with the use of the spectral tensor model, with $\sigma_U$ values measured with 80-m cup anemometer mounted on the mast. Right: comparison of lidar-based $\sigma_U$, derived from the ensemble-averaged Doppler spectrum of the CW lidar, with $\sigma_U$ values measured with 80-m cup anemometer. We show a 1:1 line for guidance, the slope of a linear regression model and the coefficient of determination $R^2$.

## 3.4 Wake detection algorithm

A wake detection algorithm is developed to determine whether the turbine is operating in free-, partial- and full-wake situations.

The algorithm relies on 10-min statistics of the lidar measurements and follows the approach of Held and Mann (2019). The idea is to detect the increase in turbulence originating from wakes with respect to the free wind conditions. This can be done by measuring turbulence intensity $TI_{LOS}$ and the relative turbulence difference measured by two lidar beams pointing at two



opposite rotor sides, $\delta TI_{LOS}$. The detection parameters are:

$$TI_{LOS} = \frac{\sigma_{LOS}}{u_{LOS}}, \qquad \delta TI_{LOS} = \frac{TI_{LOS,B1} - TI_{LOS,B2}}{\langle TI_{LOS,B1}, TI_{LOS,B2} \rangle}, \tag{14}$$

where $B1$ and $B2$ refer to the PL lidar beam notation given in Fig. 2. Due to its location (see Fig. 1), the mast is either in the wake of T04 for wind directions coming from south-west or in the wake of the upstream turbines for north-east directions.
As a consequence, we cannot rely on mast measurements to monitor free wind conditions for wind directions within our range of interest. Therefore, we propose an alternative approach, which relies on lidar measurements only. At first, we fit the wake detection parameters to a probability distribution function (pdf) using data from the wake-free wide direction sector ($97°$– $220°$). We select a log-normal and normal pdf for $TI_{LOS}$ and $\delta TI_{LOS}$, and choose the $99th$ percentile as conservative thresholds characterizing the limit of the normal range of the site-specific free wind conditions. This results in $TI_{LOS,99} = 0.276$ and $\delta TI_{LOS,99} = 0.416$. Hence, we compare the detection parameters in wake sectors to the precomputed thresholds and classify accordingly. The parameters are shown as function of the turbine yaw positions and classified as partial-wake (blue markers) and full-wake (red markers) in Fig. 5 (left and middle). A partial-wake situation is detected for $\delta TI_{LOS} > \delta TI_{LOS,99}$, whereas the sign of $\delta TI_{LOS}$ indicates which half of the rotor is affected by the wake. A full-wake is detected when both beams exhibit high turbulence $TI_{LOS,B1,B2} > TI_{LOS,99}$, but $\delta TI_{LOS} < \delta TI_{LOS,99}$. This condition appears when both beams are measuring inside the wake. Figure 5 (right) illustrates the measured fatigue blade root flapwise bending moments for wind speeds between 8 and 10 m/s, as function of the turbine orientation. Fatigue load levels are normalized with respect to the average value computed using load measurements from the free wind wide direction sector. The 10-min periods in which the turbine is operating in wake situations are shown based on the detection algorithm. A significant wake-induced effect on the load levels can be noticed. Further, we attempt to distinguish situations where the mast is in wake or in free wind, based on 10-min mast data. Turbulence from the cup at 80 m and shear derived from the cups at 57.5 m and 80 m are used as wake detection parameters (results are not shown). The wake detection results presented in Fig. 5 are obtained using the PL lidar-estimated filtered turbulence at 1.3 D. An in-depth comparison between the PL and CW lidars, filtered and unfiltered turbulence estimates and measurements at several ranges are omitted in the present work. Improved wake detection can be obtained by establishing thresholds conditional to the ambient wind conditions (i.e., wind speed, turbulence and atmospheric stability) and by assessing the detection parameters for shorter time periods (Held and Mann, 2019). More detailed detection algorithms including wake dynamic characteristics are proposed in the literature (Aitken and Lundquist, 2014; Aitken et al., 2014). Nevertheless, the proposed algorithm is able to detect 10-min periods where dominant wake effects are observed. The conservative thresholds ensure a strong wake influence in the inflow conditions and a sufficient number of 10-min periods are obtained for the purpose of load validation.

## 4 Results

The results are presented in four parts. The wake-induced effects on the reconstructed wind field parameters are analyzed in Sect. 4.1. The wind field parameters used as input for aero-elastic simulations are derived in Sect. 4.2. The one-to-one load





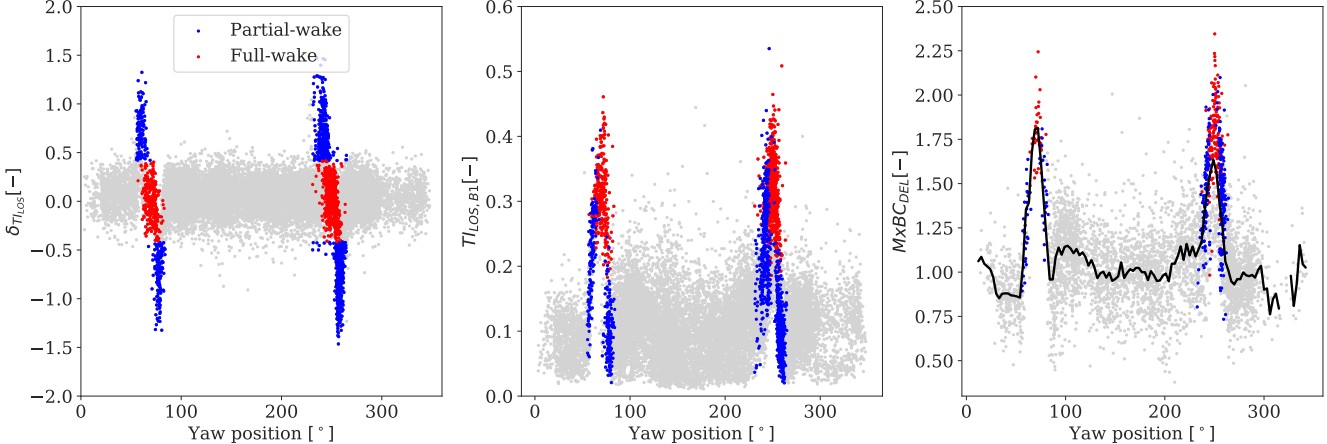

**Figure 5.** Left and middle: PL lidar-estimated 10-min wake detection parameters $\delta TI_{LOS}$ and $TI_{LOS,B1}$, as function of yaw position. Detected wake situations are shown with coloured markers: wake-free (grey), partial-wake (blue) and full-wake (red). Right: measured blade-root flapwise fatigue loads for wind speeds between 8 and 10 m/s, normalized over the average load levels in free wind conditions. The black line shows the average values binned every $3°$ directions.

comparison between simulated and measured loads and their uncertainty quantification are presented in Sect. 4.3. In Sect. 4.4, we assess the sensitivity of inflow parameters on load predictions.

## 4.1 Wake effects on reconstructed wind parameters

Wind turbine wakes lead to a region characterized by reduced wind speed and increased turbulence. We observe these effects
through the PL and CW lidar-estimated wind speed, turbulence and shear exponent in Fig. 6. Here, the slope of a linear regression model between the free wind mast-measured and lidar-estimated wind parameters in free-, partial- and full-wake situations are shown as function of the upstream distance from the rotor. The 10-min periods are classified according to the results of the wake detection algorithm in Sect. 3.4, considering south-westerly directions ($235°$– $265°$). There are 287 and 175 periods respectively, where partial- and full-wake situations are detected, while the mast is wake-free. The wake-free
mean wind speeds range between 4–14 m/s at turbulence levels between 5–15%. Although wake effects vary according to the ambient wind field, we select all measured conditions for this comparison. The influence of wakes on the lidar-estimated mean wind speed is shown in Fig. 6 (left). The reconstructed velocities in partial-wake (blue markers) and full-wake (red markers) are respectively $\sim$5% and $\sim$20% lower than ambient wind speed. The magnitude of the velocity deficit depends on the number and location of lidar beams that are measuring inside the wake. Figure 6 (left) also shows the influence of rotor induction at
shorter ranges, where low velocity is measured in the vicinity of the turbine (Mann et al., 2018). Despite that velocity recovery is expected moving downstream the wake, the induction effects are predominant. Altogether the PL and CW lidar-estimated mean wind speeds differ from each other by less than 2% in the analyzed cases. We compare lidar-measured $\sigma_u$ levels inside the wake against $\sigma_u$ measured by the mast in free wind in Fig. 6 (middle). The bias of PL lidar filtered turbulence (circle





markers) and the CW lidar filtered and unfiltered (star and triangle markers) are shown as function of upfront rotor distance. The results show clearly the increased turbulence in partial- and full-wake. The inter-comparison between PL and CW filtered turbulence in wake situations shows low bias at farther beams, where larger probe volume averaging effects are expected for the CW lidar (Dimitrov et al., 2019). The main discrepancy is found for filtered and unfiltered turbulence estimates in wake

conditions, where the latter are significantly lower. We do not observe significant induction effects on the estimated $\sigma_u$, as they affect to a much lower extent the velocity variance (Simley et al., 2016; Mann et al., 2018). A slight wake recovery can be also noticed, specifically in full-wake situations (red markers), where lower $\sigma_u$ are estimated moving downstream. The estimated shear exponent through the PL and CW lidar for free and wake conditions is shown in Fig.6 (right). As wakes expand both horizontally and vertically, wake effects can be related to a decrease of the shear exponent compared to free wind flow and

even negative values in full wake. The differences between the PL and CW lidar-estimated shear are most pronounced in full wake, where the CW measures at multiple points in the vertical direction. The fitted shear can also be used as an indicator of wake influence on inflow measurements.

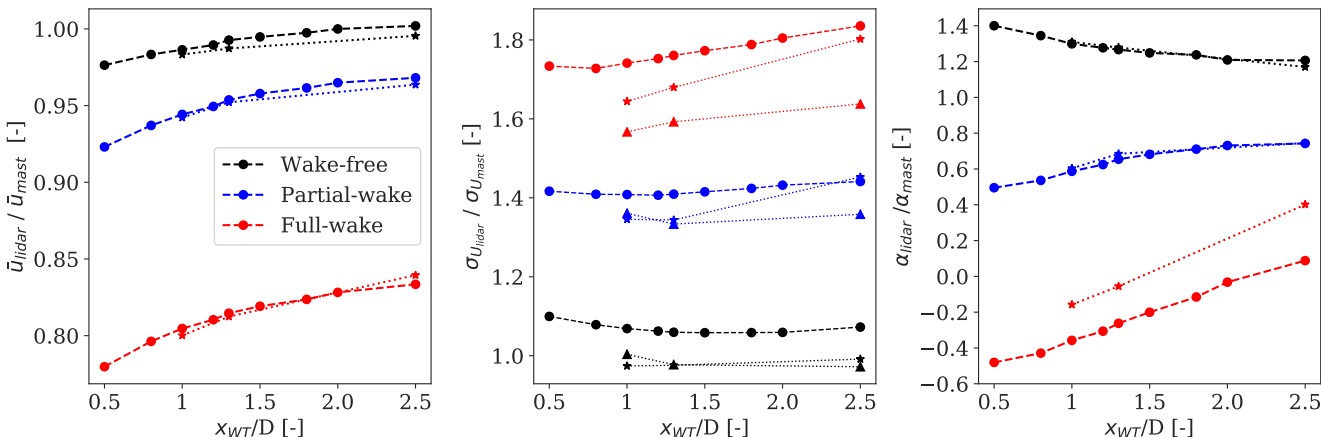

**Figure 6.** Comparison of the slope of a linear regression model between lidar-estimated and mast-measured inflow characteristics including mean wind speed (left), the standard deviation of wind speed (middle) and shear (right), when the turbine is operating in wake-free (black), partial-wake (blue) and full-wake (red) conditions and the met mast measures free wind conditions. Results are shown as function of lidar measuring distances upfront the rotor and normalized over rotor diameter. The results for the PL lidar are shown by the dash-line with circle markers, whereas those for the CW lidar by the dotted line with star markers. The triangle markers shows unfiltered turbulence obtained from the ensemble-average Doppler spectrum of the CW lidar.

In addition to the wake-induced effects on the average flow properties, turbulence spectral properties are also affected in wake regions. Earlier work on this subject showed a shift of the wake spectrum towards low length scales, compared to the free wind spectrum, both in wind tunnel and field experiments (Vermeer et al., 2003). Although large variations in length scales occurred

due to atmospheric stability, it was generally observed that wake induced turbulence is characterized by a significantly smaller length scale than that for ambient turbulence (Chamorro et al., 2012). Furthermore, wake-added turbulence can be modelled




using synthetic turbulence field with a small length scale, as done for the DWM model (Larsen et al., 2008). Based on these findings, we extract the turbulence spectra parameters of the Mann model, with focus on the length scale $L$, in free-, partial- and full-wake situations. By comparing lidar spectra to spectra from sonic anemometer in wake-free conditions at NKE, it was found that lidar measurements can qualitatively represent turbulence spectra, although differences increase for turbulence

length scales comparable to the probe volume length (Peña et al., 2017; Dimitrov et al., 2019). We ensemble-average lidar radial velocity spectra using the central beam ($B0$) of the PL lidar. Assuming that the turbine is aligned with the inflow wind direction, the central beam pointing upstream at hub height is ideally measuring the wind fluctuations of the horizontal velocity component. In this case, minimal contamination effects from other velocity components are expected. Typically, three auto-spectra of the wind velocity components as well as one-point cross-spectrum are fitted simultaneously to the theoretical spectra

to derive the Mann-model parameters. However, as we measure a single LOS spectrum, we assume $\Gamma = 3$, which is suitable for the terrain and climate for free-wake conditions (Peña et al., 2017). Although $\Gamma$ impacts load predictions, the influence of the turbulence length scale was found to be predominant (Dimitrov et al., 2017, 2018). The 10-min time series of radial velocity are classified into wind speed bins and the spectra are ensemble-averaged over each wind speed bin. Then, the parameter $L$ is fitted to the ensemble-averaged spectrum weighted on number of samples in each bin. The comparison is based on the energy

spectra of the $u$-velocity component (along-wind) in free-, partial- and full-wake situations. The measured and theoretical spectra, normalized over $\sigma_u^2$, are shown in Fig. 7 (left). The aggregated measured spectra in wakes show a shift of spectrum peak towards higher wave numbers, as expected, which indicates high energy content at low turbulence length scales. Besides, increased level of variance is observed in both partial- and full-wake compared to the values in free wind conditions. The deviations between the modelled and the measured spectra increase for wake situations. This follows from the limitations of

the Mann model, which was developed for homogeneous wind flow and near-neutral atmospheric conditions, the constraints of the adopted fitting procedure and due to the uncertainty of the lidar-measured spectra. In fact, the derived length scale values are critically affected by the probe volume filtering effects, atmospheric stability conditions, sampling frequency and measurements location in the wake region. Resulting length scales of approximately 35, 15 and 7 m are estimated, respectively, for free-, partial- and full-wake conditions. These values are used to generate synthetic turbulence fields for load simulations.

The observed magnitude of decrease in longitudinal turbulence length scale is larger than a factor of two as found in Frandsen (1996), but consistent with results reported in Thomsen and Sørensen (1998) and Madsen et al. (2010), where wake-added turbulence is characterized by length scales within the range 10–25% of the free-wind length scale. In addition, we observe a steeper slope of the wake spectra towards higher wavenumbers, which reduces the turbulence energy content within the range of rotor sampling frequencies. Small-scale turbulence is also responsible for increasing the width of the Doppler spectrum

(Branlard et al., 2013; Held and Mann, 2019). We show an example of a 10-min ensemble-average Doppler spectrum obtained from the radial velocity of the CW lidar using bins $b3$ and $b8$ (see Fig. 2 for notation) at 1.3 D in partial-wake and full-wake in Fig. 7 (middle and right). The relative free wind speed measured at the met mast is 9 m/s. It can be noticed that broadening effects are present only in $b3$ (solid blue line) in the partial-wake and in both bins (solid and dashed red lines) in full-wake conditions.





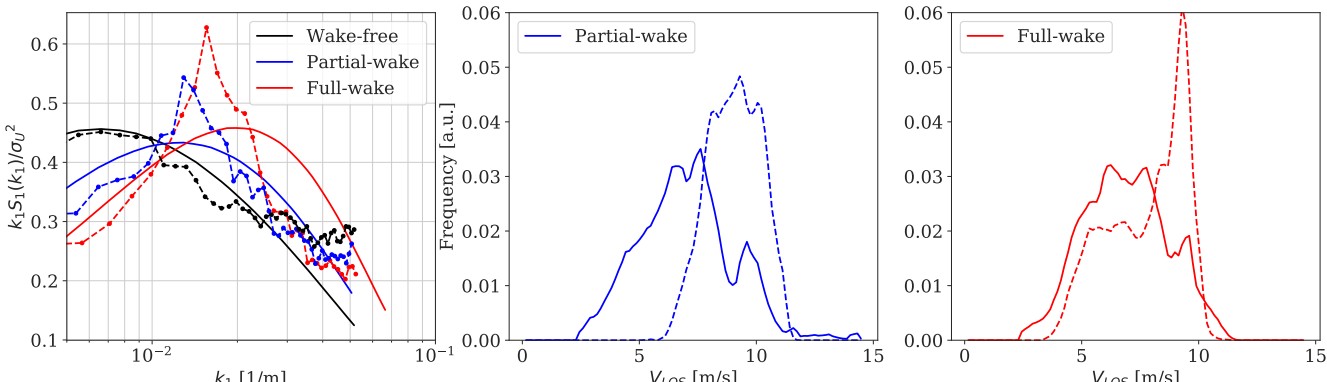

**Figure 7.** Left: comparison of the normalized ensemble-average $u_{LOS}$-spectrum based on measurements with the central beam of the PL lidar (dash-line and markers) against the fitted theoretical Mann spectra using Eq. (10) (solid line). Middle and right: normalized ensemble-average Doppler spectrum measured over a 10-min period by the CW lidar using bins *b3* (solid line) and *b8* (dash line).

## 4.2 Reconstructed inflow parameters for load simulations in wake situations

We ensure close to 500 10-min samples, distributed nearly equally among wind speed bins in the range 4–14 m/s, for free-, partial- and full-wake scenarios. We select 10-min periods within the narrow sector (97°– 109°) for free flow conditions and periods of south-west directions (235°– 265°) for wake situations. The main limitation of the current dataset is given by the
5  concurrent availability of both lidars and by the few 10-min periods at high wind speeds in full-wake situations. The comparison of the reconstructed wind field characteristics in partial-wake conditions using PL and CW lidar measurements from all ranges is presented in Fig. 8. A very good agreement can be observed for the mean wind speed; the line fits yield slopes of nearly unity and an $R^2$ of almost 100%. The filtered turbulences from the PL lidar are $\sim 2\%$ lower than those from the CW lidar. The differences can be partly explained by the larger amount of filtering occurring for farther beams of the CW lidar as well
10  as due to the distinct scanning patterns measuring an inhomogenous wind flow. When compared to the filtered turbulence, the unfiltered estimations show a significant reduction by $\sim 6\%$ (blue markers in Fig. 8 middle). A large scatter appears for the shear, veer and yaw (the latter two are not shown), which are subjected to a high level of uncertainty and highly depend on the scanning patterns. Similar results are found for full-wake situation as presented in Fig. 9. The main discrepancy is in the estimation of the shear exponent.

## 4.3 Load simulation results

The quality of load predictions ($\tilde{y}$) is evaluated through one-to-one comparison against load measurements ($\hat{y}$). Three uncertainty-related indicators are assessed, where the symbol $E(.)$ denotes the mean value and $\langle . \rangle$ the ensemble average.

- Coefficient of determination $R^2 = \langle (\tilde{y} - E(\tilde{y})^2) \rangle / \langle (\hat{y} - E(\hat{y})^2) \rangle$

- Uncertainty $X_R = \sqrt{\langle (\tilde{y}/\hat{y}) - E(\tilde{y})/E(\hat{y}) \rangle}$



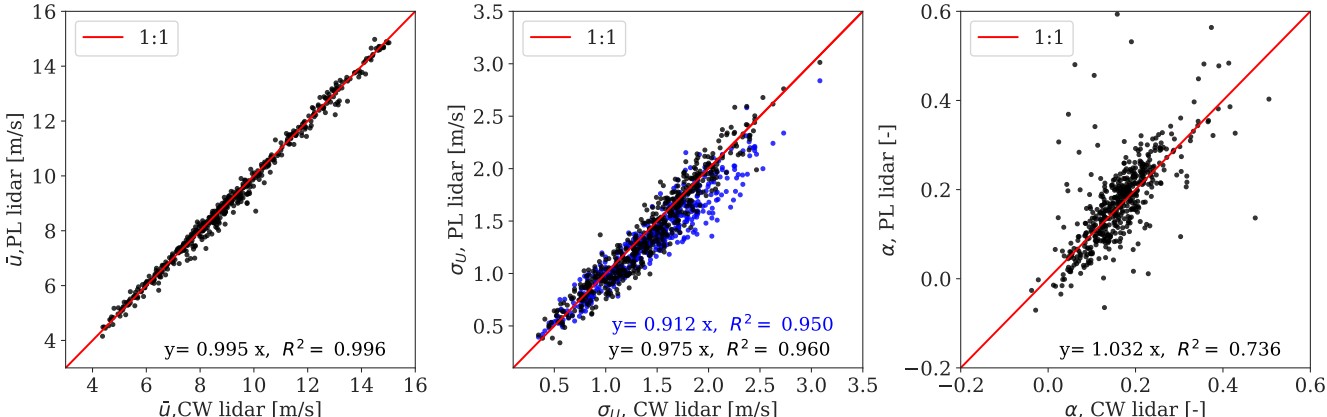

**Figure 8.** Comparison of the CW and PL lidar 10-min reconstructed wind speed (left), filtered and unfiltered turbulence in black and blue color, respectively, (middle), and shear exponent (right) for partial-wake conditions. We show a 1:1 line for guidance, the slope of a linear regression model and the coefficient of determination $R^2$.

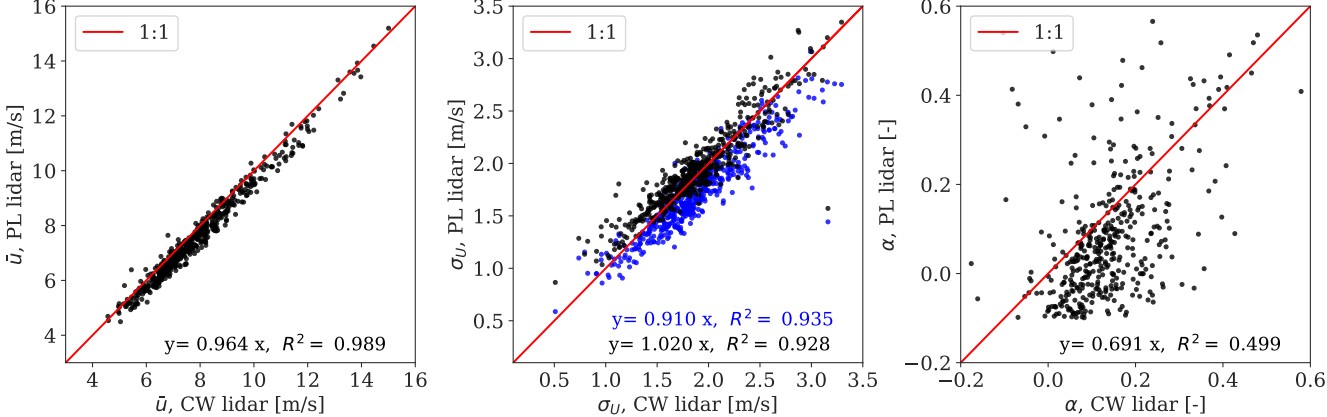

**Figure 9.** Same as Fig. 8 but for full-wake conditions

– Bias $\Delta_R = E(\tilde{y})/E(\hat{y})$

The $R^2, X_R, \Delta_R$ indicators are computed for free-, partial- and full-wake situations. The 10-min wind turbine statistics investigated hereafter include the mean power production ($\mathrm{Power_{mean}}$), the extreme loads and 1-Hz damage equivalent fatigue loads of fore-aft tower bottom bending moment ($\mathrm{M_{xTB_{max}}}, \mathrm{M_{xTB_{DEL}}}$) and flapwise bending moment at the blade root ($\mathrm{M_{xBC_{min}}}, \mathrm{M_{xBC_{DEL}}}$). Therefore, time-series of 600 s are simulated in the aero-elastic code HAWC2 and load statistics are derived. A turbulence seed with statistical properties matching those of the measured 10-min conditions is input to the load simulations. The rainflow counting algorithm is used to compute the 1-Hz damage equivalent fatigue loads with Whöler exponent of $m = 12$ for blades and $m = 4$ for the tower. The same approach is used to post-process measured loads. We derive wind



field characteristics using the PL and CW lidars as well as the mast measurements for free wind conditions. We refer to the mast-based load predictions in free wind as reference case, also denoted by $\Delta_{R,Ref}$ in the following text. A more detailed analysis is conducted for partial- and full-wake situations, where the influence of filtered and unfiltered turbulence estimates, small turbulence length scales, and wind characteristics derived from measurements at several ranges are evaluated. The predictions

uncertainties for power production and extreme loads are presented in Table 2, and for fatigue loads in Table 3. Generally, we observe lower prediction accuracy in partial- and full-wake situations compared to the reference scenario, while in some cases similar uncertainty levels are obtained. Power production levels are overestimated in partial-wake, but underestimated in full-wake within $\Delta_R/\Delta_{R,Ref} \sim 4\%$. Larger $X_R$ values are found in full-wake compared to the reference, although $R^2$ is above 96%, which indicates a good correlation. Although $\Delta_R/\Delta_{R,Ref}$ drops to 1% in partial-wake using PL lidar measure-

ments up to 1.3 D, induction effects are dominant at these ranges, leading to lower wind speed estimation. In regard to extreme loads $(\mathrm{M_{xTB_{max}}}, \mathrm{M_{xBC_{min}}})$, we obtain $\Delta_R/\Delta_{R,Ref} \sim 2\%$ in partial- and full-wake, when using unfiltered turbulence estimates and length scales extracted in free-wind conditions. Simulations based on filtered turbulence consistently overestimate extreme load levels $(\Delta_R/\Delta_{R,Ref} \sim 3\text{-}7\%)$, whereas simulations with low length scales significantly reduce extreme loads up to $\sim 7\%$ compared to reference case. Overall, higher $X_R$ values are derived in wakes compared to the reference, while $R^2$

remains above 89% in all analyzed cases. It should also be noticed that the maximum loads do not increase significantly in wake situations, since the wind speed in the wakes is lower than the free wind (Larsen et al., 2013). Fatigue load predictions in partial-wake conditions with $L = 35$ m show $\Delta_R/\Delta_{R,Ref}$ as low as 3% (see Table 3). The most significant deviations are observed for $\mathrm{M_{xTB_{DEL}}}$ and $\mathrm{M_{xBC_{DEL}}}$ in full-wake conditions. The bias of blade-root and tower-bottom predictions, for simulations based on filtered turbulence measures and $L = 35$ m, are approximately 21% higher in full-wake compared to the

reference case. Filtered turbulence estimates are predicted with the use of the spectral velocity tensor model and statistics are found to be approximately 9% higher compared to unfiltered turbulence derived from the Doppler radial velocity spectrum (see Fig. 9-middle). Correspondingly, lower fatigue load predictions are obtained using unfiltered turbulence measures, although $\Delta_R/\Delta_{R,Ref} \cong 11\%$ (see Table 3). Overall, extreme and fatigue load predictions show low uncertainty when unfiltered turbulence estimates are used as input in simulations. Furthermore, fatigue loads are found to correlate significantly better when

a synthetic turbulent field characterized by small length scales is used. This is demonstrated by lower $X_R$ for $\mathrm{M_{xTB_{DEL}}}$ and $\mathrm{M_{xBC_{DEL}}}$ compared to that resulting from simulations with a large length scale. Indeed reducing $L$ from 35 m (free-wind conditions) to 7 m (fitted in full-wake conditions) reduces extreme tower-bottom loads by 7% and fatigue blade-root loads by 15%. These results show the importance of characterizing turbulence spectral parameters for load analysis, as previously demonstrated in Thomsen and Sørensen (1998), Sathe et al. (2012), and Dimitrov et al. (2017). A decrease in the turbulence

length scale leads to a shift of the Mann spectrum towards higher wavenumbers, whereas the opposite is seen for increase length scale (Dimitrov et al., 2017). The shift of the spectrum determine the turbulence energy content corresponding to the range of frequencies of the rotor harmonics, which affects the magnitude of the loading conditions. The simulations with low length scales and unfiltered turbulence measures show improved accuracy, as $\Delta_R/\Delta_{R,Ref} < 4\%$ for $\mathrm{M_{xTB_{DEL}}}$ and $\mathrm{M_{xBC_{DEL}}}$ in full-wake conditions. The inter-comparison of uncertainties of the analyzed load sensors obtained with PL and CW lidars'

reconstructed parameters, based on same flow modelling assumptions, reveal deviations of $X_R$, $R^2$ and $\Delta_R$ within 3%.



**Table 2.** List of accuracy and uncertainty values for power and extreme load validation procedures. The marker ** indicates unfiltered turbulence obtained from the ensemble-average Doppler spectrum of the radial velocity at 1.3 D.

| Case | Sensor / ranges | Mann's length scale $L$ [m] | $\text{Power}_{\text{mean}}$ | | | $\text{MxTB}_{\text{max}}$ | | | $\text{MxBC}_{\text{min}}$ | | |
|---|---|---|---|---|---|---|---|---|---|---|---|
| | | | $R^2$ | $X_R$ | $\Delta_R$ | $R^2$ | $X_R$ | $\Delta_R$ | $R^2$ | $X_R$ | $\Delta_R$ |
| Wake-free | Mast (reference) | 35 | 0.99 | 0.09 | 1.01 | 0.97 | 0.09 | 0.97 | 0.96 | 0.09 | 0.99 |
| | PL (0.7 D - 2.5 D) | | 0.99 | 0.09 | 1.01 | 0.97 | 0.09 | 0.98 | 0.96 | 0.09 | 1.01 |
| | CW (1.0 D - 2.5 D) | | 0.99 | 0.09 | 0.98 | 0.97 | 0.09 | 0.95 | 0.97 | 0.08 | 1.01 |
| Partial-Wake | PL (0.7 D - 2.5 D) | 35 | 0.99 | 0.10 | 1.05 | 0.91 | 0.12 | 1.00 | 0.91 | 0.12 | 1.05 |
| | PL (0.7 D - 1.3 D) | | 0.99 | 0.09 | 1.02 | 0.92 | 0.11 | 0.98 | 0.92 | 0.11 | 1.02 |
| | CW (1.0 D - 2.5 D)** | | 0.99 | 0.10 | 1.03 | 0.92 | 0.11 | 0.97 | 0.92 | 0.10 | 1.01 |
| | PL (0.7 D - 2.5 D) | 15 | 0.99 | 0.09 | 1.06 | 0.91 | 0.11 | 0.97 | 0.91 | 0.11 | 1.02 |
| | CW (1.0 D - 2.5 D)** | | 0.99 | 0.10 | 1.04 | 0.92 | 0.11 | 0.96 | 0.93 | 0.10 | 0.99 |
| Full-Wake | PL (0.7 D - 2.5 D) | 35 | 0.97 | 0.19 | 0.95 | 0.92 | 0.14 | 0.99 | 0.91 | 0.12 | 1.06 |
| | CW (1.0 D - 2.5 D) | | 0.96 | 0.17 | 0.96 | 0.89 | 0.14 | 0.98 | 0.89 | 0.12 | 1.05 |
| | CW (1.0 D - 2.5 D)** | | 0.96 | 0.18 | 0.95 | 0.90 | 0.14 | 0.95 | 0.89 | 0.12 | 1.01 |
| | PL (0.7 D - 2.5 D) | 7 | 0.97 | 0.18 | 0.95 | 0.90 | 0.14 | 0.92 | 0.91 | 0.12 | 0.98 |
| | CW (1.0 D - 2.5 D) | | 0.97 | 0.15 | 0.96 | 0.92 | 0.13 | 0.91 | 0.91 | 0.10 | 0.97 |
| | CW (1.0 D - 2.5 D)** | | 0.97 | 0.16 | 0.95 | 0.91 | 0.14 | 0.89 | 0.91 | 0.11 | 0.94 |

## 4.4 Sensitivity analysis

We use a first order polynomial response surface for evaluating the sensitivity of the predictions with respect to input wind variables. We consider $\bar{u}, \sigma_U/U, \alpha, \Delta\varphi, \bar{\varphi}, L$ in the analysis. The first-order polynomials are separately fitted for free-, partial- and full-wake conditions based on the PL lidar-measured wind field parameters. We ensure close to 850 10-min samples for

each case. Besides, $L$ is assumed to randomly vary between 7 and 30 m in full-wake and between 15 and 35 m in partial-wake and free-wake situations. We normalize the input variables such that their values are scaled between zero and one to allow the sensitivity study. The obtained linear regression coefficients for $\text{Power}_{\text{mean}}, \text{M}_{\text{xBC}_{\text{DEL}}}$ and $\text{M}_{\text{xTB}_{\text{max}}}$ responses are presented in Fig. 10. The power predictions are strongly driven by the reconstructed mean wind speed at hub height as shown in Fig. 10 (left). This indicates that the observed $\Delta_R$ values are mostly explained by the uncertainty in the wind speed reconstruction.

The mean wind speed and turbulence intensity have the largest influence on the fatigue load predictions (see Fig.10-middle). In comparison to the wake-free scenario, we observe the increase effect of turbulence intensity and reduced influence of shear exponents in wake situations. This is due to the significantly high turbulence levels measured inside the wakes (up to 1.8 times higher than under free-wind conditions) and relatively low shear exponent values (see Fig. 6). The former is a well-known fatigue load driver. The latter implies small velocity gradients within the rotor area, which lead to lower blade-root fatigue

loads (Sathe et al., 2012; Dimitrov et al., 2015). The effects of $\alpha, \Delta\varphi, \bar{\varphi},$ and $L$ are secondary compared to $\bar{u}$ and $\sigma_U/U$. We





**Table 3.** Similar to Table 2 but for fatigue load validation procedures.

| Case | Sensor / ranges | Mann's length scale $L$ [m] | MxTB$_{DEL}$ $R^2$ | $X_R$ | $\Delta_R$ | MxBC$_{DEL}$ $R^2$ | $X_R$ | $\Delta_R$ |
|---|---|---|---|---|---|---|---|---|
| Wake-free | Mast (reference) | 35 | 0.86 | 0.19 | 0.93 | 0.84 | 0.22 | 1.01 |
| | PL (0.7 D - 2.5 D) | | 0.85 | 0.20 | 0.97 | 0.83 | 0.23 | 1.09 |
| | CW (1.0 D - 2.5 D) | | 0.86 | 0.18 | 0.91 | 0.84 | 0.21 | 1.01 |
| Partial Wake | PL (0.7 D - 2.5 D) | 35 | 0.81 | 0.18 | 0.95 | 0.83 | 0.23 | 1.04 |
| | PL (0.7 D - 1.3 D) | | 0.82 | 0.18 | 0.93 | 0.83 | 0.22 | 1.02 |
| | CW (1.0 D - 2.5 D)** | | 0.80 | 0.17 | 0.92 | 0.85 | 0.19 | 1.00 |
| | PL (0.7 D - 2.5 D) | 15 | 0.83 | 0.17 | 0.94 | 0.83 | 0.21 | 0.98 |
| | CW (1.0 D - 2.5 D)** | | 0.83 | 0.16 | 0.90 | 0.86 | 0.17 | 0.94 |
| Full Wake | PL (0.7 D - 2.5 D) | 35 | 0.78 | 0.19 | 1.11 | 0.84 | 0.24 | 1.22 |
| | CW (1.0 D - 2.5 D) | | 0.74 | 0.18 | 1.08 | 0.81 | 0.20 | 1.19 |
| | CW (1.0 D - 2.5 D)** | | 0.73 | 0.17 | 1.01 | 0.80 | 0.19 | 1.12 |
| | PL (0.7 D - 2.5 D) | 7 | 0.82 | 0.15 | 1.09 | 0.85 | 0.18 | 1.07 |
| | CW (1.0 D - 2.5 D) | | 0.79 | 0.16 | 1.05 | 0.84 | 0.16 | 1.02 |
| | CW (1.0 D - 2.5 D)** | | 0.79 | 0.16 | 0.97 | 0.84 | 0.15 | 0.97 |

observe slightly higher sensitivity of $L$ in full-wake compared to partial-wake and free-wind conditions. However, according to the results in Table 3, the length scale parameter has a significant impact on loads when assessed independently. Finally, $\bar{u}$ and $\sigma_U/U$ have the largest influence on the extreme tower bottom loads in Fig. 10 (right). Overall, the order of importance of the analyzed inflow parameters are comparable with the more detailed sensitivity studies provided in Dimitrov et al. (2018).

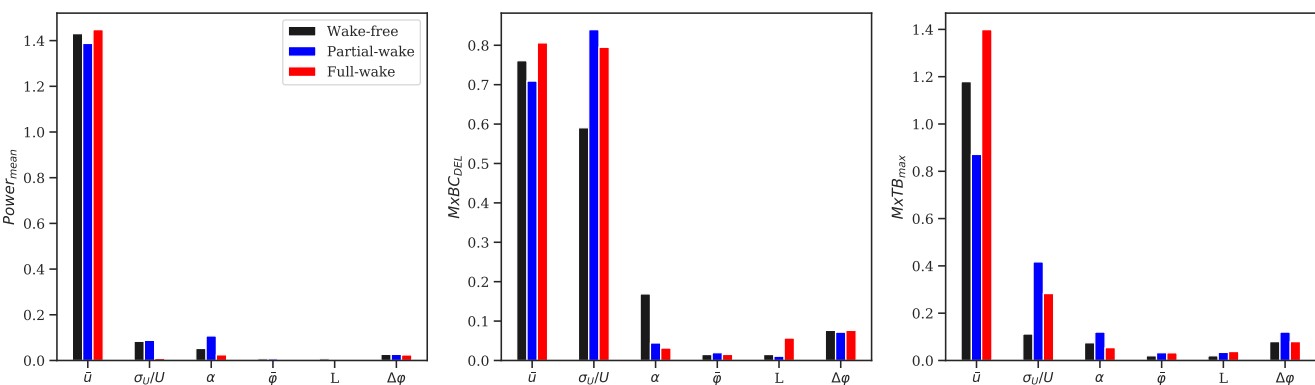

**Figure 10.** Regression coefficients of a linear response surface model identifying the sensitivity of wind field parameters on selected load sensors.





We provide detailed scatter plots of measured and predicted load sensors used in the analysis in Figs. A1-A5 in the Appendix. We analyze the bias and uncertainty of $\mathrm{Power_{mean}}, \mathrm{M_{xBC_{DEL}}}, \mathrm{M_{xTB_{max}}}$ predictions with respect to the inflow wind speed in Fig. 11. We observe larger deviations of the selected sensors at low wind speeds, which gradually decrease for higher winds. The deviations in the reference case (black line) and wake situations are a combination of uncertainty in the reconstructed wind

profiles, aero-elastic model uncertainty, load measurement uncertainty as well as statistical uncertainty (Dimitrov et al., 2019). Although there is not sufficient information to distinguish among the various uncertainty sources, we assume that the deviations are due to the error in the wind field representation only. The power predictions uncertainties with respect to mean wind speed in free-, partial- and full-wake situations are plotted in Fig. 11 (left). We observe a consistent overprediction of power levels in partial-wake conditions (blue line) and underprediction in full-wake conditions (red line) for the full range of wind speeds. The

predictions of $\mathrm{M_{xBC_{DEL}}}$ with respect to the mean wind speeds in free- and full-wake conditions are plotted in Fig.11 (middle). The predictions based on the unfiltered turbulence (green line) show better agreement to the reference compared to results based on filtered turbulence (red line). It is also found that largest deviations occur at low wind speeds ($\bar{u} < 8$ m/s). Finally, we show the results using unfiltered turbulence and low length scale (purple line), which provide the lowest error. The residual deviations can be partly explained by the uncertainty in turbulence statistics and spectral properties representation. Figure 11

(right) shows that comparable deviations are obtained for tower extreme loads in partial- and full-wake situations as for the reference case.

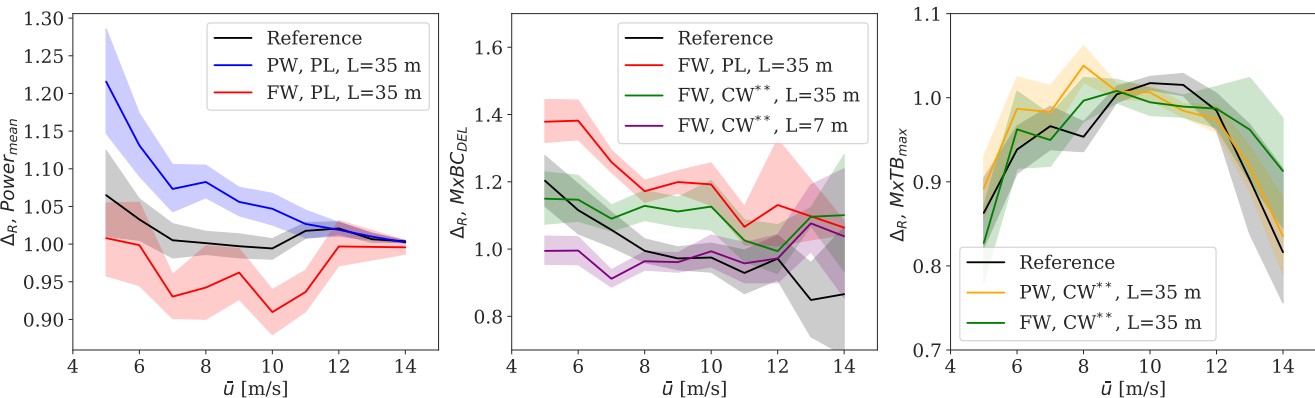

**Figure 11.** Comparison of bias (solid line) and uncertainty (error band) of selected load sensors with respect to inflow mean wind speed. The analyzed cases are shown with coloured lines in each sub-plot. The reference case denotes the mast-based free wind scenario; PW refers to partial-wake and FW refers to full-wake conditions. We show results from both PL and CW lidars and different turbulence length scales $L$. The marker ** indicates unfiltered turbulence obtained from the ensemble-average Doppler spectrum of the radial velocity at 1.3 D.



## 5   Discussion

Wind field parameters used as inputs for aero-elastic simulations are derived from PL and CW lidar measurements of the wake field behind an operating wind turbine. Although the two lidars follow different scanning patterns and the wake flow field is strongly inhomogeneous, we find a very good agreement between the PL and CW lidar-estimated horizontal wind speed and

filtered turbulence in partial- and full-wake situations. The estimation of the wind veer, yaw error and shear exponent using nacelle-mounted lidars is prone to high level of uncertainty and is affected by the scanning patterns. This is demonstrated by the larger scatter between the estimated parameters by the PL and CW lidars in wake compared to free-wind conditions (not shown). However, we demonstrate that the influence of these parameters on the loads and power predictions is minor compared to mean wind speed, turbulence intensity and length scale. Although the present work does not focus in details on the

performance of the two lidar systems, the findings indicate that the main sources of uncertainty in load predictions are related to flow modelling assumptions. Power production levels are highly dependent on the estimated mean wind speed at hub height. The observed power prediction's deviations, in both partial- and full-wake situations, indicate an inaccurate reconstruction of the mean wind speed. More precisely, the flow modelling assumptions, including horizontal homogeneous wind flow, power law vertical wind profile and linear veer within the scanned areas, introduce larger errors in wake than wake-free conditions.

Furthermore, we do not distinguish situations where the lidar beams are partly measuring inside the wake and partially outside to reconstruct the inflow wind field. This could be resolved if wake characteristics as shape, depth and center position are integrated in the WFR techniques (Trujillo et al., 2011). Deriving a rotor equivalent wind speed model, which accounts for velocity gradients as well as wake characteristics is necessary to improve the accuracy of power and load predictions. It is also important to point out that, in the region of high velocity gradients as wake edges, the probe volume averaging effects can lead

to significant errors in the estimation of the mean wind speed (Lundquist et al., 2015; Meyer Forsting et al., 2017).

It is well-established that fatigue loads are dominated by turbulence levels. However, to extract turbulence parameters by combining a turbulence model with a model of the spatial radial velocity averaging of the lidars introduces significant uncertainty under wake conditions. We describe the wake flow as a homogeneous field by using the Mann spectral tensor model fitted using PL lidar measurements at hub height. Nonetheless, wake fields are highly inhomogeneous and spectral properties

vary significantly within the rotor region (Kumer et al., 2017). We demonstrate the importance of characterizing turbulence spectra for the load analysis. However, a detailed analysis including atmospheric stability effects on turbulence spectra and wake characteristics can potentially reduce the uncertainty of load predictions (Sathe et al., 2012; Dimitrov et al., 2017). Furthermore, cross-contamination effects and probe volume averaging effects become larger in wakes, as the size of turbulence eddies decrease to length scales comparable or lower than the lidar probe volume. These effects increase the uncertainty of

extracted turbulence length scales. Despite this, fatigue load predictions show significant improvement by using low turbulent length scales; spectral analysis of measured and predicted loads is required for a better understanding of the accuracy of the lidar-fitted synthetic turbulence field in wakes.

We demonstrate that improved fatigue load predictions are obtained using unfiltered turbulence measures from the Doppler radial velocity spectrum. However, the estimation of $\sigma_U^2$ from the $\sigma_{LOS}^2$ relies on flow homogeneity and Taylor's frozen





turbulence hypothesis (Taylor, 1938). These assumptions are sound for large-scale wind fluctuations and free flow over flat and homogeneous terrain, but not valid in wakes (Schlipf et al., 2010). The current wind field modelling approach omits the large-scale meandering of wakes, which has strong impact on power and load predictions (Larsen et al., 2013). These uncertainty sources, among others, can partially explain the observed deviations.

The influence of wake effects on power and load levels depend on the wind farm layout, ambient wind speed and turbulence, and atmospheric stratification, among others. The current state-of-the-art approach to predict wake flows and their influence on wind turbine operations relies on engineering-like wake models (Frandsen, 2007; Madsen et al., 2010). These models ensure an acceptable level of accuracy, robustness and computational cost. Previous studies carried out load validation using the effective turbulence model and the DWM model, which are recommended in the IEC 61400-1. The inter-comparison of the two models

showed fatigue load prediction deviations of 20% (Thomsen et al., 2007). The effective turbulence approach under predicted fatigue load levels at spacings larger than 5 D (Schmidt et al., 2011). Although the DWM model showed a very fine agreement between power and load predictions in wake conditions (Larsen et al., 2013), these studies did not quantify uncertainty in a systematic approach. Recently, Reinwardt et al. (2018) estimated fatigue load biases in the range 11–15% for the tower bottom and 8–21% for the flapwise bending blade using the DWM. Results from earlier studies show deviations of the same or even

larger order of magnitude compared to the results from our load validation approach. Despite of the discussed shortcomings, load validation under wake conditions based on lidar measurements may be already a viable alternative to the engineering wake models. We will soon evaluate whether the differences in the calculated loads using lidar-estimated wind characteristics in wakes are larger compared to the uncertainties in the load calculations with state-of-the-art wake models such as DWM.

## 6 Conclusions

We demonstrated a procedure for carrying out load validation in partial- and full-wake conditions using measurements from two types of forward-looking nacelle lidars: a pulsed and continuous wave system. The suggested procedure characterized wake-induced effects by means of wind field parameters commonly used as input for load simulations. These parameters were reconstructed using lidar measurements of the wake flow field. We considered the uncertainty of load predictions in wake-free sectors using mast-measured free wind conditions as the reference case. We quantified the uncertainty and bias of power and

load predictions in partial- and full-wake conditions using lidar-estimated wind field characteristics. The reconstructed mean wind speed, turbulence intensity as well as the turbulence length scale in wake conditions were found to be the most influential parameters on the predictions. Power production levels under wake conditions were strongly related to the mean wind speed at hub height, whereas the vertical and horizontal wind profiles had negligible effects on those levels. Power predictions in partial- and full-wake conditions deviated up to 4% compared to the reference case. Fatigue loads were affected by turbulence

characteristics inside the wake. The use of a spectral velocity tensor model to derive turbulence parameters introduced significant uncertainty under wake conditions. The tower-bottom and blade-root bending moments predictions deviated by 3% in partial-wake conditions, and were overestimated by 21% in full-wake conditions using filtered turbulence measures and turbulence length scales typical of free-wind conditions. The simulation bias in full-wake conditions was reduced to 11% us-





ing unfiltered turbulence measures derived from the ensemble-average Doppler radial velocity spectrum. The measured and predicted fatigue and extreme loads were found to correlate significantly better when a synthetic turbulent field characterized by a low turbulence length scale was used. Furthermore, low turbulence length scales led to a strong reduction of load levels, reducing the bias of fatigue loads to 4% under wake conditions. However, estimating turbulence characteristics under wake

5  conditions using measurements from nacelle-mounted lidars was prone to high level of uncertainty due to probe volume effects and flow modelling assumptions. We demonstrated the applicability of nacelle-mounted lidar measurements to extend load and power validations under wake conditions and highlighted the main challenges. Further investigation is necessary to verify that the observed uncertainty of predictions are comparable with results using state-of-the-art wake models recommended by the IEC standard.

10  **Appendix A: Figures with load statistics comparisons**

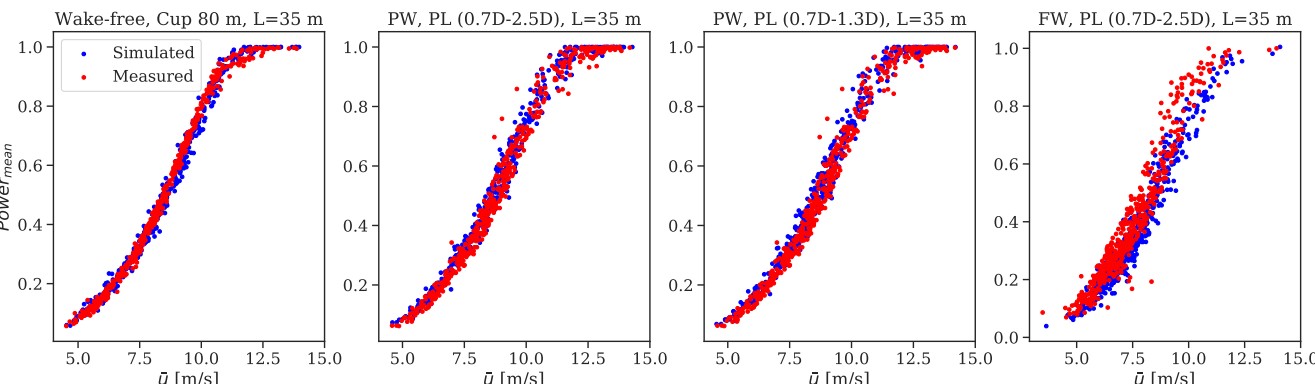

**Figure A1.** Scatter plots of the normalized measured and predicted power mean realizations used in the analysis.

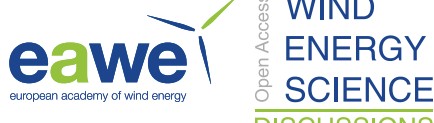

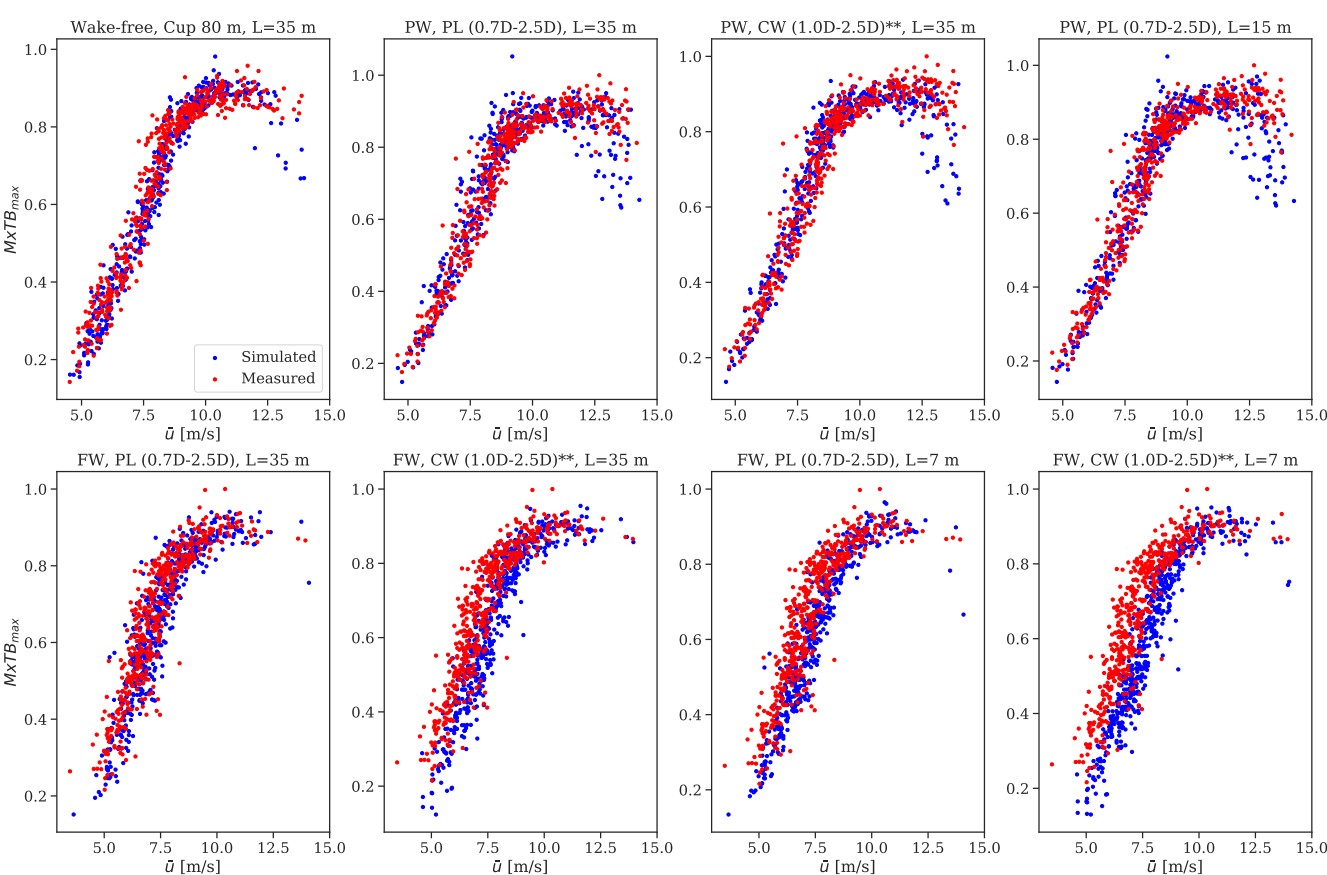

**Figure A2.** Scatter plots of the normalized measured and predicted extreme fore-aft tower bottom bending moment realizations used in the analysis.



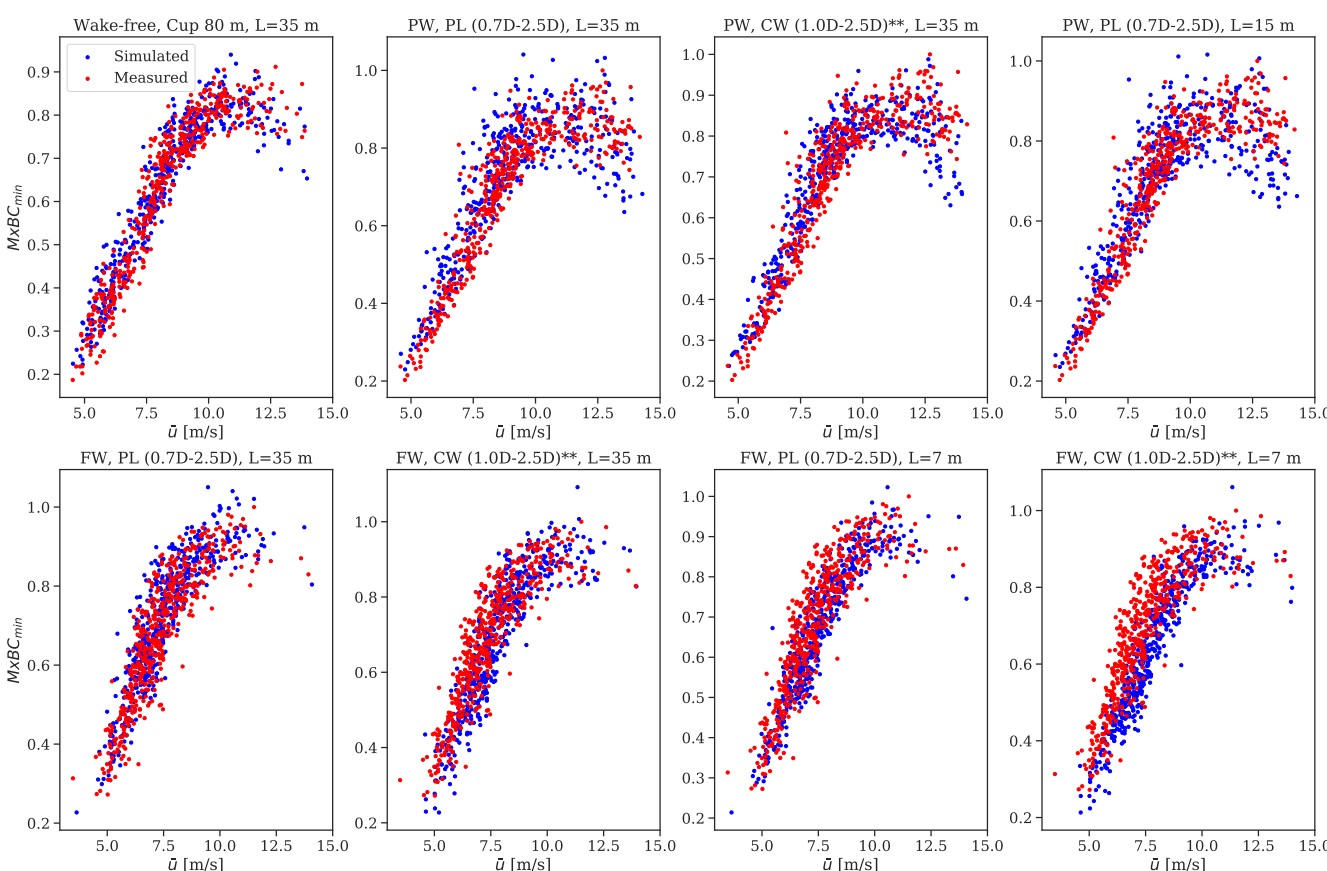

**Figure A3.** Scatter plots of the normalized measured and predicted extreme flapwise bending moment at the blade root realizations used in the analysis.





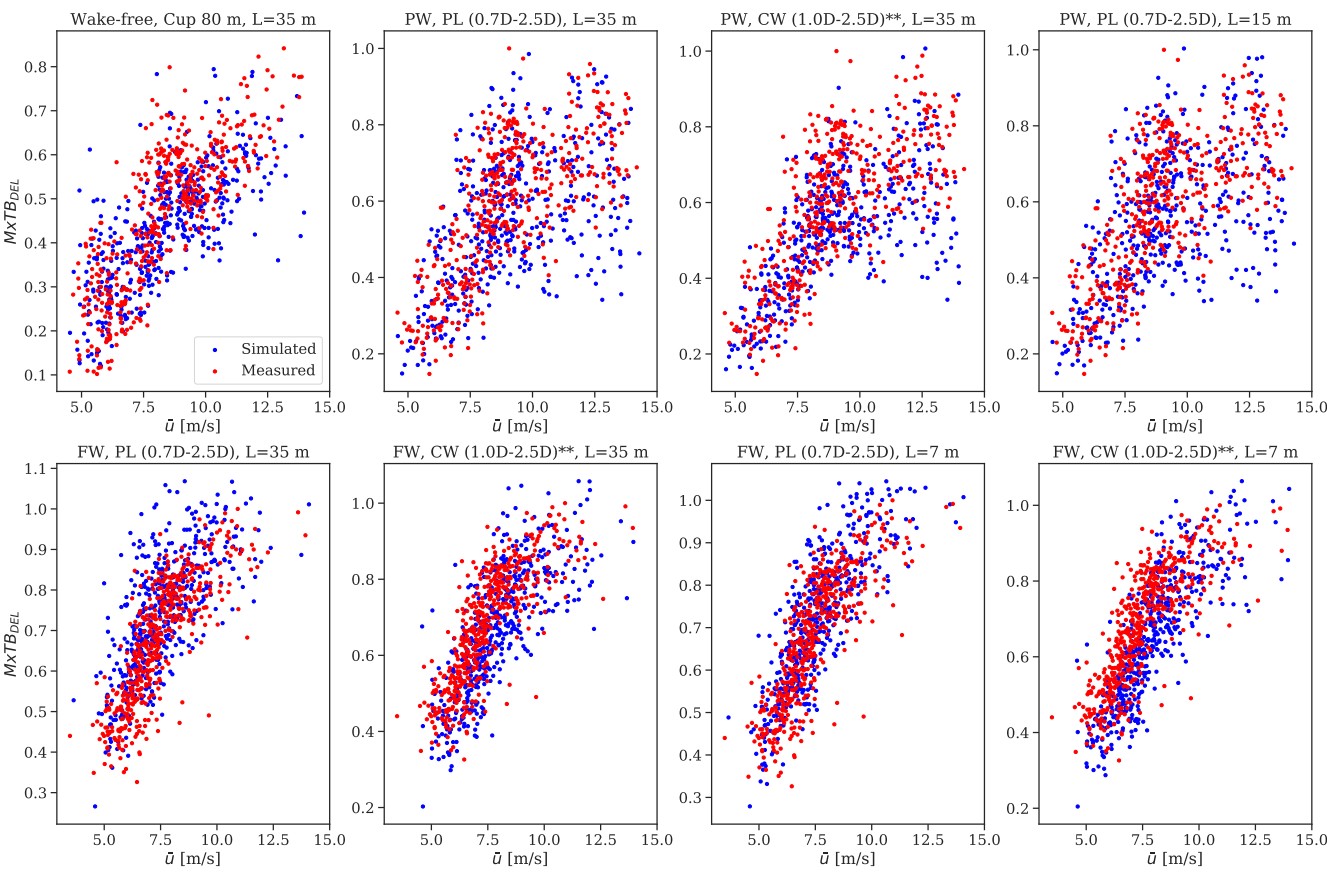

**Figure A4.** Scatter plots of the normalized measured and predicted fatigue fore-aft tower bottom bending moment realizations used in the analysis.

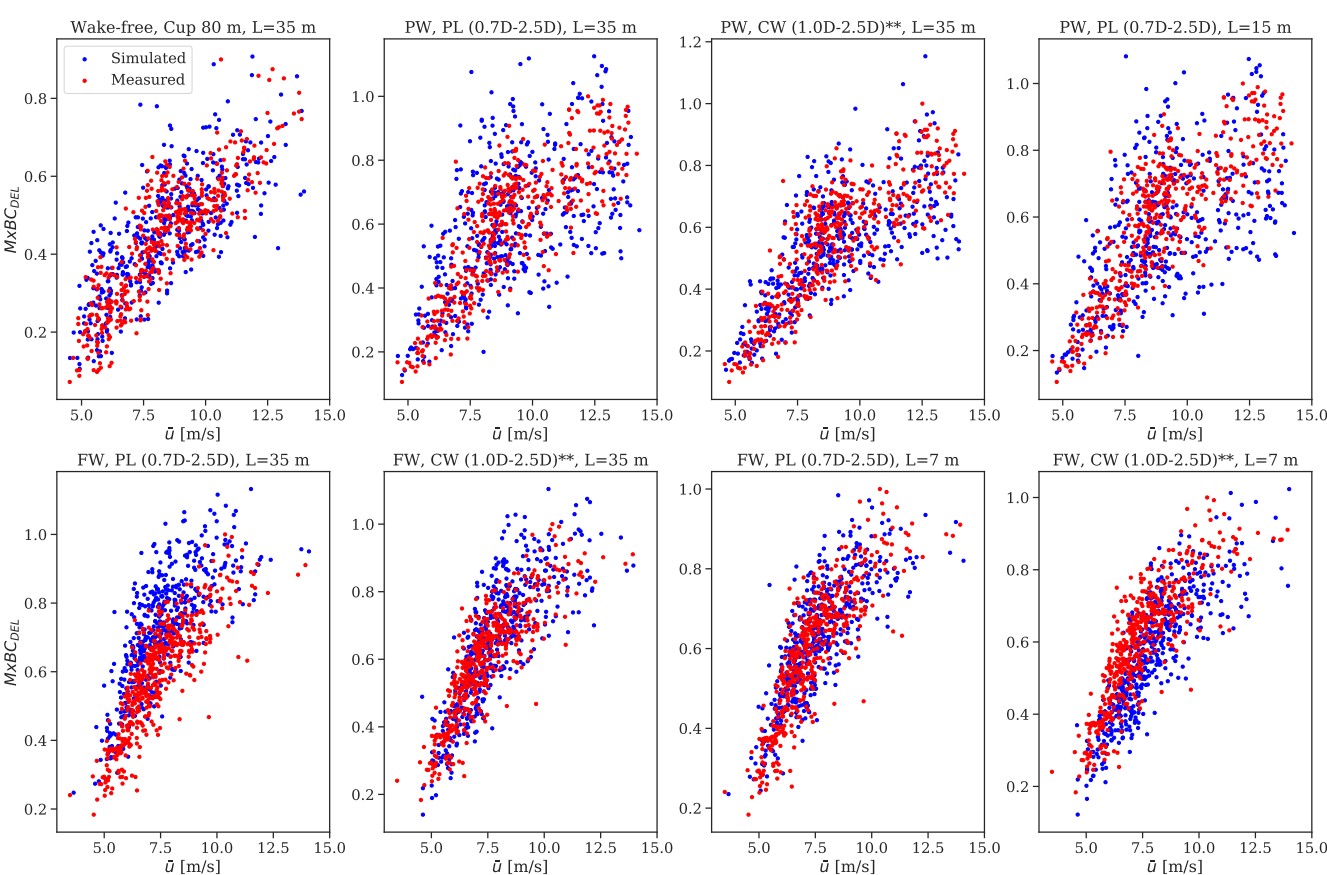

**Figure A5.** Scatter plots of the normalized measured and predicted fatigue flapwise bending moment at the blade root realizations used in the analysis.



*Data availability.* Turbine data are not publicly available because there is a non-disclosure agreement between the partners in the UniTTe project. Lidar and mast data can be requested from Rozenn Wagner at DTU Wind Energy (rozn@dtu.dk).

*Competing interests.* The authors declare that they have no conflict of interest.

*Acknowledgements.* Special thanks to Rozenn Wagner for making the NKE dataset available for our study. Thanks to the lidar manufacturers

5   from Avent and ZephIR for providing their systems. Thanks to Vattenfall and Siemens Gamesa Renewable Energy for providing the site and turbine to conduct the measurement campaign



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
