# Peer review of "Aero-elastic load validation in wake conditions using nacelle-mounted lidar measurements"

_Wind Energy Science, 2020_

## Referee Comment (RC1) · Anonymous Referee #1 · 22 Mar 2020

This is an interesting and thorough paper about the use of nacelle lidar-derived wind conditions for load validation in waked conditions. The paper builds on the work of Dimitrov et al. 2019 "Wind turbine load validation using lidar‐based wind retrievals" to extend the load validation procedure from freestream conditions to waked operation. This is a relevant area of work because of the uncertainty that exists in current methods for performing load validation in waked operation, relying on wake models, as opposed to site-specific measurements. Opportunities to improve the validation of loads in waked conditions are therefore of great interest. Overall, the paper is well-written, but I have a couple general comments and many mostly minor comments that I feel should be addressed.

[Figure]

-More motivation for the presented work should be included. For example, I was wondering how accurate current IEC-recommended methods for estimating loads in waked conditions are, and if there is a clear need to improve these methods using lidar-measured wind conditions. Additionally, I was wondering why the authors did not compare the accuracy of the load predictions to the accuracy when using these IEC-standard wake modeling methods. It was not until the end of the paper (last paragraph of Section 5) that these topics were discussed. I believe this material should be moved to the introduction to better motivate the research and to explain the scope of the current work (i.e., that comparison with DWM, etc. is not part of the current work).

-Many of the paragraphs throughout the manuscript are very long, often an entire page. Organizing them into smaller paragraphs would greatly improve the readability. For example, pg. 2, pg. 11, pg. 14, pg. 17.

Specific comments

1) Pg. 1, ln. 10: "...lead to an increase of the relative error as low as 4%" Kind of a confusing sentence. Maybe something like "only increase the relative error by 4% in some cases"?

2) Pg. 2, ln. 2: "To account for these effects, aero-elastic load simulations are combined with wake models..." How are lifetime fatigue loads calculated? Are the fatigue loads with and without the added wake turbulence/wake models added together, weighted by the frequency of occurrence for waked and freestream conditions?

3) Pg. 2: The lidar literature review is very concentrated on the activities of DTU Wind Energy. DTU is certainly one of the leaders in lidar for wind energy, but including more works from other organizations would make a more representative review. As an example, a couple other relevant references are:

-Turbulence: Newman, J. F. and Clifton, A.: An error reduction algorithm to improve lidar turbulence estimates for wind energy, Wind Energy Science, 2017.

-Wakes: Iungo, G.V.; Porté-Agel, F. Volumetric Lidar Scanning of Wind Turbine Wakes under Convective and Neutral Atmospheric Stability Regimes. J. Atmos. Ocean. Technol. 2014.

4) Table 1: Can you explain the difference between u_hub (mean wind speed at hub height) and U (in the TI definition)? Or are these the same and could be written with the same symbol?

5) Section 3.1: A figure showing the coordinate system and variables would be helpful.

6) Eq. 4+5: Please explain the meaning of a 3D LOS vector. The component along the lidar beam direction makes sense, but what are the other two components?

7) Eq. 6: Should C_ind only be applied to the first term in the brackets, aligned with the rotor orientation (the direction of the rotor thrust)? This is how it is written in Dimitrov et al. 2019. Additionally, a little more information about C_ind would be useful for the reader. What are the input parameters to C_ind?

8) Pg. 7, ln. 17: "The parameters (u_hub,..." Because of the induction zone model, I imagine the induction factor is also a parameter that is estimated. Is this true?

9) Pg. 7, ln. 28: "The velocity fluctuations, denoted by u = (u,v,w), are expected homogenous..." Shouldn't it be that the "statistics" of the velocity fluctuations are expected homogeneous?

10) Eq. 7: Should R be a function of the separation vector r?

11) Eq. 8: R_i,j should be a function of r here, not k.

12) Pg. 9, ln. 24: "The relation between the covariance matrix of the LOS is then..." It seems something is missing here. The relationship between the covariance matrix and what else?

13) Eq. 13: I believe this equation results in a matrix. How do you go from a matrix to a scalar value used to scale the LOS variance?

14) Pg. 10, lns. 4-6: The full details can be left to Pena et al. 2017, but a little more detail about how the different beam directions are combined to find the u-component variance would be appreciated, since "computing the variance of Eq. (5)" is hard to interpret.

15) Pg. 10, ln. 8: "The filtered turbulence derived from CW and PL lidars are plotted..." Which beams are used to derive the turbulence?

16) Pg. 11, ln. 27: "The conservative thresholds ensure a strong wake influence in the inflow conditions..." This seems like a good approach for this study, but using the algorithm for a full load validation would probably miss some of the more benign wake conditions. Would this lead to overestimating the wake loads if only the strong wake conditions are simulated?

17) Pg. 12, ln. 5: "...through the PL and CW lidar-estimated wind speed, turbulence and shear exponent in Fig. 6." Is Eq. 6 used to find the wind parameters, or another method? Since Eq. 6 combines multiple ranges together to estimate the wind field parameters, I'm guessing you are using a different approach here.

18) Pg. 12, ln. 9: "while the mast is wake-free": How are you determining if the mast is wake free? For example, from Fig. 1, at 265 degrees, the mast looks like it will be waked.

19) Pg. 13, ln. 3: "...shows low bias at farther beams": Please clarify what you mean by low bias in this case. Bias between the two types of lidars?

20) Pg. 14, ln. 29: "Small-scale turbulence is also responsible for increasing the width of the Doppler spectrum." Can you explain how the turbulence scale impacts the Doppler spectrum width? I would expect this to be a function of the standard deviation, but it isn't clear how the length scale directly impacts this.

21) Pg. 14, ln. 32: "It can be noticed that broadening effects are present only in b3..." Where is the evidence of broadening effects? Wouldn't this require the velocity

spectrum in freestream conditions for comparison?

22) Pg. 15, ln. 18: Please compare with the coefficient of determination equation in Dimitrov et al. 2019. It appears there are some typos in the equation listed here.

23) Pg. 16, ln. 5: Why is only M_XBC_min investigated, as opposed to M_XBC_max?

24) Pg. 17, ln. 8: "Delta_R / Delta_R,Ref": Should this be "Delta_R - Delta_R,Ref"? Or "Delta_R / Delta_R,Ref = 104%"?

25) Pg. 17, ln. 10: "induction effects are dominant at these ranges..." Doesn't the WFR in Eq. 6 already account for induction effects to estimate the freestream (or in this case wake) wind speed?

26) Pg. 17, ln. 17: "...as low as 3%": Should this be "as high as 3%"? This seems to be the highest % over Delta_R,Ref, not the lowest.

27) Pg. 18, ln. 7: "The obtained linear regression coefficients for Power..." This is a nice analysis. Are the trends the same for M_XBC_min and M_XTB_DEL (which aren't shown)?

28) Table 3: It would probably be worth repeating the (short) caption from Table 2 here.

29) Pg. 19, ln. 1: "...slightly higher sensitivity of L in full-wake compared to partial-wake and free-wind conditions." The partial wake loads still show a strong dependence on L (although less than full wake). Any thoughts on why the regression coefficient for L is almost zero for partial wake?

30) Pg. 23, ln. 7: "Further investigation is necessary to verify that the observed uncertainty of predictions are comparable with results using state-of-the-art wake models..." How would this be investigated? Would you need freestream measurements of the inflow conditions to use as inputs to the wake model? If so, how would the freestream conditions be measured, given that the mast appears to be waked for much of the sector where the turbine is waked?

---

## Referee Comment (RC2) · Anonymous Referee #2 · 24 Mar 2020

Review of "Aero-elastic load validation in wake conditions using nacelle-mounted lidar measurements"

**Summary:** Upstream measurements of nacelle-mounted Doppler lidars are used to characterize the inflow wind field of a wind turbine and to set-up simulations for load and power predictions. These predictions are then validated by
(i) quantifying uncertainty indicators between lidar-based prediction and measurements of on-board sensor in waked conditions
(ii) comparing uncertainty indicators quantified between mast-based prediction and on-board sensors in free wind conditions with the results of (i).
Lastly, the sensitivity of the results to the input parameters of the load simulation is investigated. It is concluded that using lidars for load and power validations is a viable possibility, but further research is needed to compare with IEC standards.

**General comments:** The manuscript motivates the research question and its relevancy. The methods describe the measurement site, the scan set-up of the lidars, and their processing in sufficient detail, but is scant with information on the simulation and the on-board sensors. My main issues with manuscript are (explained in more detail in the specific comments below):

1. The same inflow parameters that are used to characterize a free-stream inflow are applied to a waked inflow without modification. This includes the assumption of a power-law wind profile that is not valid within a wake.
2. Section 4.3, which presents the validation, is missing structure and components of the validation are not defined. Further, it is not clear to me, why the mast was chosen for the wake-free reference case and not lidar, because this adds another variable in the interpretation.
3. The on-board sensor and the simulations are introduced in only one sentence with a reference to another paper. Since they are as integral as the lidars for the validation, they should receive more attention in my opinion.

I classified this as major revisions, because the first issue could change the results or modify the research questions and conclusions, and the other comments point at an incomplete manuscript.

**Language:** I noticed only a few typos or grammar mistakes in the manuscript with the disclaimer that I am not a native English speaker. Parts of the manuscript would benefit from structuring with subsections and paragraphs (sections 1, 3.4, 4.1, and 4.3 specifically).

**Specific comments**

- Page 1, line 8-9: This sentence reads to me, that lidar-based load predictions in waked conditions are compared against the mast-based predictions from a mast located in the free wind at the same time. But I believe that the intended meaning is, that the uncertainty of lidar-based predictions against on-board sensors in waked conditions are compared with the uncertainty of mast-based predictions against sensor data in free wind conditions.

- Page 1, line 10: Why is only the smallest increase of the relative error given? What was the largest observed increase?

- Page 1, lines 10-11: How do they impact the predictions? (e.g. do low wind speed lead to a better uncertainty of the prediction or the opposite?)

- Fig. 2: The remainder of the manuscript often uses x/D as the upwind distance relative to the turbine with the lidars (opposite to here, where it is the downwind distance from the turbine that

causes the wake).

- Page 5, eq. (1): Within the wake, the wind speed profile is not following a power law profile. Therefore, I am not convinced that the shear exponent resulting from a partially or fully waked inflow is meaning full. Since the inflow parameters are later used as input for the simulations, the simulated conditions can be expected to be very different from the conditions experienced by the wind turbine.
For the CW lidar it seems possible to retrieve spanwise fields of the longitudinal mean velocity provided with some interpolation. Is it possible to initialize the simulations with them instead? As it is, the approach would be better suited to answer what errors are entailed by applying procedures developed for free stream conditions to waked conditions.

- Page 6, line 2: The simulations should be introduced in more detail.

- Page 7, line 10: I assume that the assumption of homogeneity is referring to horizontal homogeneity only and not including vertical homogeneity?

- Fig. 4: Why is the cup anemometer and not the sonic anemometer used for the turbulence measurements? I would expect that the standard deviation from a sonic anemometer is better since it is not affect by cross-contamination and inertia.

- Eq. (14): Why are only the positions B1 and B2 in the upper half of the rotor considered and not the beams B3 and B4 in the lower half?

- Page 12, line 9: How was it determined that the mast is wake-free?

- Fig. 6: As I understood from section 3.1, the effects of the induction zone were removed from the lidar measurements in the wind field reconstruction and the text mentions that the left panel shows reconstructed mean wind speeds (page 12, line 12). Therefore, I am wondering why the effects of the induction zone are present in the lidar data or whether the shown data is based on eq. (5) and not eq. (6)?

- Fig. 6: I am confused, because the axis labels state the ratio between lidar measurements and mast measurements, but the caption states the slope of a linear regression between.

- Page 14, lines 12-14: From the text I understand that for each wind speed bin an ensemble-averaged spectrum was computed and to each of those spectra the model is fitted to estimate L (i.e. for each wind speed bin a separate L is computed). However, that does not line up with single spectrum presented in Fig. 7 (left) and three length scales reported in line 24.

- Page 14, lines 17-18: Since the shown spectra are normalized with their respective variance, I don't see this from Fig. 7 (left) and only from Fig. 6 (middle).

- Page 14, line 29: At a given wave number, the turbulence kinetic energy depends on the absolute value of energy spectrum and not its slope. Therefore, I am not understanding this sentence. Also, the term rotor sampling frequency could be explained, because I could not find it in quick search and I am not familiar with it.

- Page 15, line 2: How many bins do you have or what is lowest amount of samples in a bin?

- Page 15, line 16 to page 16, line 8: The overall validation approach seems sensible to me. However, in my opinion, it is not clearly written down and has to be pieced together from two

different places in the manuscript. Separate subsections for the power and the two bending moment might be help to make it easier to digest.

In particular, precise definitions of $\tilde{y}$ and $\hat{y}$ were not provided in this manuscript (I looked at the definitions and explanations in Dimitrov et al. (2019) and hope they are also valid here). Also, the generation of two separate bin averaged wind speed ensemble from the previous section should be recalled here. The difference between the mean and the ensemble average with respect to this data set should be explained explicitly. I believe that an extensive rework of section 3.4 is needed, because I misunderstood an essential part of the validation on my first reading and I believe that was not entirely my own fault.

- Page 15, line 18: Check equation. The square should be outside of the bracket.

- Page 15, line 19: Check equation. Missing a square.

- Page 17, line 8: I am not yet understanding why this bias ratio was chosen as an indicator for the behavior of the uncertainty. With a mast-based reference case, there are two influencing factors with (1) the differences between mast vs. lidar and (2) waked vs. free wind conditions. Why not use lidar-based predictions for free wind conditions as the reference? Then only waked vs. free wind conditions remain, which would make interpretation easier.

Also, in later occurrences of the bias ratio it is not stated whether it is an over- or underestimation (I am assuming that both 1.02 and 0.98 would be given as 2%).

- Page 17, line 20: The section is already quite loaded. It might be worth to consider to separate the filtered and unfiltered comparison from the rest of the section. The same might be considered for the length scale.

- Page 18, lines 11: I believe it should be "the increased effect".

- Page 18, lines 12: I believe it should be "higher".

- Page 18, line 15: Why is the effect of L in the sensitivity analysis minor, but it had an effect of up to 15% in the previous section (Table 2)?

- Page 22, lines 10-11: Consider rephrasing this sentence, because it can be understood in two ways.

- Page 22, lines 27-28: Reiterating a previous comment, I am not convinced that the power law should be applied within the wake, because the wind profile has a different shape. The conclusion regarding the vertical wind profile here is based on the sensitivity to the shear exponent and I am not convinced that it holds for the same reason, because the simulated wind fields might have been very different from the real wind field.

Also, I am confused regarding the horizontal wind profile, because I cannot remember that horizontal gradients were accounted for in the wind field retrieval. Is that referring to the wind veer maybe?

- Page 29, lines 23-24: The reference of Dimitrov et al. (2019) seems to be out of place and should appear after Dimitrov et al. (2018) assuming the sorting hierarchy is first author alphabetical followed by year.

---

## Author Comment (AC1) · 3 May 2020

**Author response to reviewer 1**

The authors response is shown in red

Changes implemented in the new version of the manuscript are in blue

We thank the reviewer for the comments, which we consider very important. Here our response to each of them.

More motivation for the presented work should be included. For example, I was wondering how accurate current IEC-recommended methods for estimating loads in waked conditions are, and if there is a clear need to improve these methods using lidar-measured wind conditions. Additionally, I was wondering why the authors did not compare the accuracy of the load predictions to the accuracy when using these IEC standard wake modeling methods. It was not until the end of the paper (last paragraph of Section 5) that these topics were discussed. I believe this material should be moved to the introduction to better motivate the research and to explain the scope of the current work (i.e., that comparison with DWM, etc. is not part of the current work).

Following the comment of the reviewer, we now provide further motivation on the need to improve load validation approaches in wakes, and clarify the scope of the work along the manuscript. Further, "Sect. 2.1 Requirements for load validation in wakes" is fully dedicated to define the scope of the current work and introduce the reader with the elements of the load analysis.

**The following text is added on page 2 line 9:**

The comparison of fatigue loads predicted using the DWM and the effective turbulence approach by the IEC showed a discrepancy of 20% (Thomsen et al., 2007). The uncertainty varied according to the inflow conditions and spacing between turbines. The work of Larsen et al. (2013) showed a very fine agreement between both power and load measurements and predictions based on a site-specific calibrated DWM model for the Dutch Egmond aan Zee wind farm. However, the study did not quantify uncertainty in a systematic approach. More recently, Reinwardt et al. (2018) estimated fatigue load biases in the range 11–15% for the tower bottom and 8–21% for the blade-root flapwise bending moments using the DWM. To date, these approaches are characterized by a significant level of uncertainty, due to the stochastic nature of environmental conditions and the various simplifying assumptions used in the wake model definitions (Schmidt et al., 2011). Further, these results motivate the need for improving wind turbine load validation approaches in wake conditions.

The following replacement is done in the abstract on page 1 line 4:

From: "We account for wake-induced effects by means of wind field parameters commonly used as inputs for load simulations, which are reconstructed using lidar measurements."

To: "The wake flow fields are described by lidar-estimated wind field characteristics, which are commonly used as inputs for load simulations, without employing wake deficit models."

The following text is added in the introduction on page 2 line 21:

Based on these findings, we extend the load validation procedure defined in Dimitrov et al. (2019) to include wake conditions. Therefore, wake-induced effects are accounted for by means of

wind field parameters commonly used as inputs for load simulations, which are reconstructed using lidar measurements, yet without employing wake deficit models.

-Many of the paragraphs throughout the manuscript are very long, often an entire page. Organizing them into smaller paragraphs would greatly improve the readability. For example, pg. 2, pg. 11, pg. 14, pg. 17.

To improve readability, we restructured the manuscript as following:

- Section 4.1 Wake effects on reconstructed wind parameters
- Section 4.2 Wake effects on turbulence spectra properties

We renamed section 4.3 from "Load simulation results" to "Load validation procedure" and re-arranged the section 4.3 as:

- Section 4.3 Load validation procedure
- Section 4.3.1 Power predictions
- Section 4.3.2 Extreme load predictions
- Section 4.3.3 Fatigue load predictions

"Sect. 4.4 Sensitivity analysis" is now divided in:

- Sect. 4.4 Sensitivity of inflow parameters on load predictions

- Sect. 4.4.1 Uncertainty distribution as function of wind speed

Specific comments:

1) Pg. 1, ln. 10: ": : :lead to an increase of the relative error as low as 4%" Kind of a confusing sentence. Maybe something like "only increase the relative error by 4% in some cases"?

The following text is modified in the abstract on page 1 lines (10)

From: "Compared to the reference case, the simulations in wake conditions lead to an increase of the relative error as low as 4%"

To: "Compared to the free-wind case, the simulations in wake conditions lead to increased relative errors (4-11%)."

2) Pg. 2, ln. 2: "To account for these effects, aero-elastic load simulations are combined with wake models: : :" How are lifetime fatigue loads calculated? Are the fatigue loads with and without the added wake turbulence/wake models added together, weighted by the frequency of occurrence for waked and freestream conditions?

In our opinion the current best practice is to derive lifetime fatigue loads based on site-specific conditions, e.g., using mast measurements, if available, or assuming appropriate probability distribution of the wind conditions at the site. With knowledge of the ambient conditions and the wind direction distribution in relation to the wind farm layout, the lifetime fatigue loads can be derived by weighting the frequency of occurrence of waked and free-stream conditions. Here, we derive the 1-Hz damage equivalent load for each 10-min period, without computing lifetime fatigue loads. However, the proposed approach could also provide information on the frequency of occurrence of waked and free-stream conditions and it can be extended to derive lifetime fatigue loads.

3) Pg. 2: The lidar literature review is very concentrated on the activities of DTU Wind Energy. DTU is certainly one of the leaders in lidar for wind energy, but including more works from other organizations would make a more representative review. As an example, a couple other relevant references are:

-Turbulence: Newman, J. F. and Clifton, A.: An error reduction algorithm to improve lidar turbulence estimates for wind energy, Wind Energy Science, 2017.

-Wakes: Iungo, G.V.; Porté-Agel, F. Volumetric Lidar Scanning of Wind Turbine Wakes under Convective and Neutral Atmospheric Stability Regimes. J. Atmos. Ocean. Technol. 2014.

The recommended references are now added.

4) Table 1: Can you explain the difference between  $u_{hub}$  (mean wind speed at hub height) and U (in the TI definition)? Or are these the same and could be written with the same symbol?

We now replace the U with u in Table 1 and throughout the manuscript.

5) Section 3.1: A figure showing the coordinate system and variables would be helpful.

The coordinate system and orientation of the axis can be derived from Fig. 2 and the definition of the angle is self-explanatory in the text. As the manuscript is already quite long, we added a figure showing the coordinate system and the variables in the appendix.

6) Eq. 4+5: Please explain the meaning of a 3D LOS vector. The component along the lidar beam direction makes sense, but what are the other two components?

The  $T_{LOS}$  is a 3x3 rotational matrix obtained from the product of two rotations about the yand z-axes,  $\psi_y$  and  $\psi_z$  (see figure above). The 1st row of  $T_{LOS}$  defines the transformation between the LOS (along the beam of the lidar) and the reference-coordinate system (x,y,z) of figure above. Theoretically, the 2nd and 3rd rows of  $T_{LOS}$  would refer to the transformation between two vectors perpendicular to the LOS and the reference system. As the lidar measures only the LOS, then only the 1st row of  $T_{LOS}$  is used.

In order to clarify this, we correct the manuscript and specify that only the 1st row of  $T_{LOS}$  is used before introducing Eq. 5.

The following text is added on page 7 line 8 before Eq. 5:

As lidars measure only the LOS velocity, the first row alone of  $T_{LOS}$  is considered. The relation between the wind vector and the LOS velocity is expressed in terms of matrix transformations as:

$$\boldsymbol{u_{los}} = \boldsymbol{T_{LOS}}\boldsymbol{T_1}\boldsymbol{u}.$$

7) Eq. 6: Should  $C_{ind}$  only be applied to the first term in the brackets, aligned with the rotor orientation (the direction of the rotor thrust)? This is how it is written in Dimitrov et al. 2019. Additionally, a little more information about  $C_{ind}$  would be useful for the reader. What are the input parameters to  $C_{ind}$ ?

Eq. 6 is now corrected.

The following text is added on page 7 line 16:

The two-dimensional induction model assumes longitudinal and radial variation of the induced wind velocity. The resulting induction factor  $C_{ind}$  is computed as:

$$C_{ind} = \left[1 - a_0 \left(1 - \frac{\xi_x}{\sqrt{1 + \xi_x^2}}\right) \cdot \left(\frac{2}{\exp\left(+\beta_a \epsilon_a\right) + \exp\left(-\beta_a \epsilon_a\right)}\right)^2\right],\tag{2}$$

where  $a_0$  is the induction factor at the rotor center area;  $\xi_x = x/R_{rotor}$  is the distance from the rotor normalized by the rotor radius;  $\rho_a = \sqrt{y^2 + z^2}/R_{rotor}$  is the radial distance from the rotor center axis;  $\epsilon_a = \rho_a/\sqrt{\lambda_a(\eta_a + \xi_x^2)}$ , where  $\gamma_a = 1.1$ ,  $\beta_a = \sqrt{2}$ ,  $\alpha_a = 8/9$ ,  $\lambda_a = 0.587$ ,  $\eta_a = 1.32$  (Dimitrov et al., 2019).

8) Pg. 7, ln. 17: "The parameters  $(u_{hub},::::$ " Because of the induction zone model, I imagine the induction factor is also a parameter that is estimated. Is this true?

This is correct, the  $C_{ind}$  is now included in the sentence.

9) Pg. 7, ln. 28: "The velocity fluctuations, denoted by u = (u,v,w), are expected homogenous: : "Shouldn't it be that the "statistics" of the velocity fluctuations are expected homogeneous?

We corrected and added "statistics" in the sentence.

10) Eq. 7: Should R be a function of the separation vector r?

R is defined as the covariance tensor. R(r) is a function of the separation vector when computing a two-point turbulent statistics, where the selected two points are separated by a vector r. However, for single-point turbulent statistics r=0, the covariance tensor (R(r=0) = R) can be expressed in matrix form that contains variance and covariance terms of the wind field u=(u,v,w).

To clarify, the following text is replaced on page 7 line 28:

From: "The velocity fluctuations, denoted by u = (u,v,w), are expected homogeneous in space (Mann, 1994). The covariance tensor of single-point turbulent statistics can be written as:"

To: "The velocity fluctuations (u', v', w'), where (') denotes fluctuations around the mean value, are expected homogeneous in space (Mann, 1994). It follows that the auto- or cross-covariance function between two points can be defined only in terms of the separation distance as  $R_{ij}(\mathbf{r}) =$  $u'_i(\mathbf{x})u'_j(\mathbf{x} + \mathbf{r})\rangle$ , where i, j = (1,2,3) are the indices corresponding to the components of the wind field,  $\langle \rangle$  denotes ensemble averaging, and  $\mathbf{r} = (r_1, r_2, r_3)$  is the separation vector in the threedimensional Cartesian coordinate system. The covariance tensor of single-point turbulent statistics  $(\mathbf{R}(\mathbf{r} = 0) = \mathbf{R})$  can be written as:

$$\boldsymbol{R} = \begin{bmatrix} \langle u'u' \rangle & \langle u'v' \rangle & \langle u'w' \rangle \\ \langle v'u' \rangle & \langle v'v' \rangle & \langle v'w' \rangle \\ \langle w'u' \rangle & \langle w'v' \rangle & \langle w'w' \rangle \end{bmatrix} = \begin{bmatrix} \sigma_u^2 & \sigma_{uv} & \sigma_{uw} \\ \sigma_{vu} & \sigma_v^2 & \sigma_{vw} \\ \sigma_{wu} & \sigma_{wv} & \sigma_w^2 \end{bmatrix},$$
(3)

**11) Eq. 8: $R_i$ , j should be a function of r here, not k.**

**This is now corrected.**

12) Pg. 9, ln. 24: "The relation between the covariance matrix of the LOS is then: : :" It seems something is missing here. The relationship between the covariance matrix and what else?

**The following text is replaced on page 9 line 24:**

From: "The relation between the covariance matrix of the LOS is then expressed in terms of  $\sigma_u^2$  as:"

To: "The relation between the covariance matrix of the LOS components and that of the undisturbed wind field is then expressed in terms of  $\sigma_u^2$  as:"

13) Eq. 13: I believe this equation results in a matrix. How do you go from a matrix to a scalar value used to scale the LOS variance?

 $R_{LOS}$  is the full covariance matrix containing three vector components: the LOS and the other two vectors perpendicular to the LOS. However, as lidar measures only the LOS component, only the first component of  $R_{LOS}$  is measured. Therefore the ratio in Eq. 13 defines the relation between the LOS variance (as scalar) and the variance of the u-component.

The following text is replaced on page 9 line 26:

From: "Since only LOS velocities are measured by the nacelle-mounted lidar, the ratio in Eq. (13) identifies the relation between the LOS variance and the wind field variance in the longitudinal

direction"

To: "Note that  $R_{LOS}$  is expressed as a full covariance matrix containing three vector components. However, as only LOS velocities are measured by the nacelle-mounted lidar, only the first component of  $R_{LOS}$  is measured. It follows that the ratio in Eq. (14) identifies the relation between the LOS variance and the wind field variance in the longitudinal direction"

14) Pg. 10, lns. 4-6: The full details can be left to Pena et al. 2017, but a little more detail about how the different beam directions are combined to find the u-component variance would be appreciated, since "computing the variance of Eq. (5)" is hard to interpret.

We added the equation below, which shows the relation between the LOS variance and the variance of the u-component. The equation is derived from the extended form of Eq. 5 by applying the variance operator. In its extended form, as shown in Peña et al. 2017, the LOS variance and the lidar orientation angles are known variables, while the variances  $\langle u'u' \rangle, \langle v'v' \rangle, \langle w'w' \rangle$  and covariance  $\langle u'w' \rangle$  are unknown. As the measurements of the CW lidar are grouped in 10 bins within each range, we require minimum four of these bins to solve the system and derive the variance of the u-component. This is valid under the assumption of homogenous turbulence.

The following text is now added on Page 10 lines 4-6:

By assuming homogeneous turbulence, we use the scanning pattern to account for cross-contamination of different velocity components and extract 10-min  $\sigma_u^2$  statistics by computing the variance of Eq. (5) as:

 $Var(u_{los}) = Var((\cos\psi_y\cos\psi_z\cos\varphi - \cos\psi_y\sin\psi_z\sin\varphi)u - (\cos\psi_y\cos\psi_z\sin\varphi + \cos\psi_y\sin\psi_z\cos\varphi)v + (\sin\varphi)w)$ (4)
By solving the variance operator and neglecting the resulting terms  $\langle u'v' \rangle$  and  $\langle v'w' \rangle$ , as exby solving the variance operator and neglecting the resulting terms  $\langle u v \rangle$  and  $\langle v w \rangle$ , as explained above,  $\sigma_u^2$  is derived as shown in Eq. (10) in Peña et al. (2017).

15) Pg. 10, ln. 8: "The filtered turbulence derived from CW and PL lidars are plotted: : :" Which beams are used to derive the turbulence?

To derive the filtered turbulence estimates from both the CW and PL lidars, we used all the beams and ranges. To derive the unfiltered turbulence from the CW lidar, we use only measurements at 1.3 D.

We replaced the text in the manuscript on page 10 line 8:

From: "The filtered turbulence derived from the CW and PL lidars are plotted respectively in Fig. 4 (left and middle), whereas the unfiltered turbulence derived from the CW lidar is shown in Fig. 4 (right)."

To: "The filtered turbulence derived from CW and PL lidars, using all the ranges and beams, are plotted respectively in Fig. 4 (left and middle), whereas the unfiltered turbulence derived from the CW lidar measurements at 1.3 D are shown in Fig. 4 (right)."

16) Pg. 11, ln. 27: "The conservative thresholds ensure a strong wake influence in the inflow conditions: : :" This seems like a good approach for this study, but using the algorithm for a full load validation would probably miss some of the more benign wake conditions. Would this lead to overestimating the wake loads if only the strong wake conditions are simulated?

The conservative thresholds ensure the strong impact of wakes on the 10-min measured/simulated wind field and wind turbine statistics. As we run the uncertainty quantification separately for partial-, and full-wake conditions, this approach allows us to identify and differentiate the conditions within the 10-min periods. Further, as the uncertainty increases for wake situations compared to the free case scenario, we indeed expect these results to be on the conservative side.

17) Pg. 12, ln. 5: ": : :through the PL and CW lidar-estimated wind speed, turbulence and shear exponent in Fig. 6." Is Eq. 6 used to find the wind parameters, or another method? Since Eq. 6 combines multiple ranges together to estimate the wind field parameters, I'm guessing you are using a different approach here.

We corrected and rephrased the text. In the analysis shown in Fig. 6, we investigate how the lidar-based wind characteristics varies as function of the upstream ranges in free, partial- and full-wake situations. For this particular analysis, we disregard induction effects.

The following text is now added on page 12, line 5:

From: "We observe these effects through the PL and CW lidar-estimated wind speed, turbulence and shear exponent in Fig. 6. Here, the slope of a linear regression model between the free wind mast-measured and lidar-estimated wind parameters in free-, partial- and full-wake situations are shown as function of the upstream distance from the rotor."

To: "Here, the slope (m) of a linear regression model between the free wind mast-measured and lidar-estimated wind parameters in free-, partial- and full-wake situations is shown. In this particular analysis, the lidar-based wind parameters are derived from Eq. (6) evaluated a different upstream distances from the rotor without including induction effects."

18) Pg. 12, ln. 9: "while the mast is wake-free": How are you determining if the mast is wake free? For example, from Fig. 1, at 265 degrees, the mast looks like it will be waked.

Some details were described on page 11 line 19. However, we now add an Appendix to describe the application of the wake detection algorithm to the mast measurements.

Appendix A: Wake detection from mast measurements

The wake detection algorithm (see Sect.3.4) is extended to the mast measurements to classify 10-min periods where the mast is in free or wake situations. For this purpose, turbulence observations from the cup anemometer at 80 m and vertical wind shear computed using the measurements from the cup anemometers at 57.5 and 80 m are used as wake detection parameters. Their 99th percentiles are used as conservative thresholds to characterize the limits of the normal range of the site-specific free wind conditions. The resulting thresholds are  $TI_{mast,99} = 0.20$  and  $\alpha_{mast,99} = -0.02$ . If one of the two limits is exceeded within a 10-min period, the mast is considered in wake conditions and shown with green markers in Fig. 1.

Figure 1: Left and middle: 10-min observations of the turbulence intensity and vertical wind shear at the mast as function of turbine yaw position. Free wind conditions relative to the mast are identified with grey markers, and waked situations with green markers. Right: PL-estimated 10min wake detection parameter  $TI_{LOS,B1}$ . Detected wake situations of turbine T04 are shown with coloured markers: wake-free (grey), partial-wake (blue) and full-wake (red). The 10-min periods, where the mast is affected by wakes are shown in green markers.

19) Pg. 13, ln. 3: ": : :shows low bias at farther beams": Please clarify what you mean by low bias in this case. Bias between the two types of lidars?

The following text is replaced on page 13 line 3:

From: "The inter-comparison between PL and CW filtered turbulence in wake situations shows low bias at farther beams, where larger probe volume averaging effects are expected for the CW lidar (Dimitrov et al., 2019)."

To: "The difference between PL and CW filtered turbulence in wake situations (circle and star markers) decreases at farther beams, where larger probe volume averaging effects are expected for the CW lidar (Dimitrov et al., 2019)."

20) Pg. 14, ln. 29: "Small-scale turbulence is also responsible for increasing the width of the Doppler spectrum." Can you explain how the turbulence scale impacts the Doppler spectrum width? I would expect this to be a function of the standard deviation, but it isn't clear how the length scale directly impacts this.

The small-scale (high frequency) fluctuations will be filtered out by the probe volume of lidar measurements, especially if the characteristic turbulence length scale is lower than the probe volume size. However, the analysis of the Doppler radial velocity spectrum can recover information of the turbulence at the different scales. It is expected that small-scale turbulence from wakes result in broadening of the Doppler spectrum compared to ambient flow conditions. The widening of the Doppler spectrum relates to increased variance (or standard deviation) of the LOS velocity. These effects were reported in Held and Mann (2019).

21) Pg. 14, ln. 32: "It can be noticed that broadening effects are present only in b3: : :" Where is the evidence of broadening effects? Wouldn't this require the velocity spectrum in freestream conditions for comparison?

---

## Author Comment (AC2) · 3 May 2020

**Author response to reviewer 2**

The authors response is shown in red
Changes implemented in the new version of the manuscript are shown in blue

We thank the reviewer for the comments, which we consider very important. Here our response to each of them.

Summary: Upstream measurements of nacelle-mounted Doppler lidars are used to characterize the inflow wind field of a wind turbine and to set-up simulations for load and power predictions. These predictions are then validated by:
(i) quantifying uncertainty indicators between lidar-based prediction and measurements of onboard sensor in waked conditions
(ii) comparing uncertainty indicators quantified between mast-based prediction and on-board sensors in free wind conditions with the results of (i).
Lastly, the sensitivity of the results to the input parameters of the load simulation is investigated. It is concluded that using lidars for load and power validations is a viable possibility, but further research is needed to compare with IEC standards.

General comments: The manuscript motivates the research question and its relevancy. The methods describe the measurement site, the scan set-up of the lidars, and their processing in sufficient detail, but is scant with information on the simulation and the on-board sensors. My main issues with manuscript are (explained in more detail in the specific comments below):

1. The same inflow parameters that are used to characterize a free-stream inflow are applied to a waked inflow without modification. This includes the assumption of a power-law wind profile that is not valid within a wake.

We agree that the power law is not valid in wake conditions. The vertical wind speed gradient in wakes can be seen as the combination of the atmospheric shear, which can typically be described by a power-law profile, and the wake deficit. However, due to the complexities of wakes, i.e., horizontal and vertical meander and, at NKE in particular, scenarios with multiple wakes the velocity gradients are very complex. It would be very complicated to separate each effect from lidar measurements, without tracking the wakes' center locations in time and without knowing the wake-free atmospheric shear. With the current setup, the derived wind characteristics anyway minimize the error between the modelled wind field and lidar measurements. Fitting a power law on these data is equivalent to applying a low-order approximation to a complex nonlinear function, effectively there is minimal difference between using the average value and the power law. As shown from the the sensitivity analysis in Fig. 10, the shear exponent does not have a significant impact on the predictions. The results from Table 2 and Table 3 as well as those in Fig. 10 and Fig. 11 indicate a higher load predictions uncertainty in wakes compared to free wind conditions. As we assume that the observed deviations in load predictions are solely due to the error in the wind field representation (see Page 3 line 20), it can be inferred that the increased uncertainty is due to the flow modelling assumptions of the used wind field model. This was stated and discussed in the discussion section (see Page 21 line 9-19). In addition, we have made changes in multiple relevant parts of the manuscript, to outline the caveats of using a power law together with wake deficits.

We think that an improved scope of the work is required as well as an extended discussion

regarding the limitations due to the modelling assumptions. We therefore replaced or added text:

Page 1 line 4:

[revised manuscript text omitted]

2. Section 4.3, which presents the validation, is missing structure and components of the validation are not defined. Further, it is not clear to me, why the mast was chosen for the wake-free reference case and not lidar, because this adds another variable in the interpretation.

We conducted an extensive re-structuring of Section 4.3; more details are given in the specific comments below. The mast is here the reference because it would be unfair to choose as reference any of the two lidars. Further, the current IEC load validation approaches under free-wind conditions are based on measurements obtained from a met mast; lidars are still not recommended for load validation purposes. However, following the suggestion of the reviewer, we now use the lidars as the reference case. The load statistics in wake conditions obtained with one of the lidars are compared to the respective statistics of the same lidar in free wind conditions. We keep the results based on the mast for completeness and show the uncertainty of lidar-based load validation in free-wind conditions.

3. The on-board sensor and the simulations are introduced in only one sentence with a reference to another paper. Since they are as integral as the lidars for the validation, they should receive more attention in my opinion.

We added additional details on the on-board sensors in "Sect. 2.2. Measuring campaign" by including the location of the strain gauges on the wind turbine structures (i.e., blades and tower) and the sampling frequency of the load data.

The following text was added on page 4 line 1:

The wind turbine T04 was instrumented with sensors for load measurements at the roots of two blades, tower top, and tower bottom (Vignaroli and Kock, 2016). The strain gauges were installed at 1.5 m from the blade root flange, at 11.85 m below the lower surface of the tower top flange, and at 5.9 m above the upper surface of tower bottom flange. The data acquisition software was set to sample at 35 Hz on all channels. Additional data were provided by the supervisory control and data acquisition (SCADA) system including nacelle wind speed and orientation, power output, blade pitch angles, and generator speed.

We added additional details on the simulations in "Sect. 3 Methodology", including the description of the HAWC2 model and add relative references.

The following text was added on page 6 line 1:

Load simulations are carried out using the state-of-the-art aero-elastic HAWC2 software (Larsen and Hansen, 2007). The structural part of the code is based on a multi-body formulation assembled with linear anisotropic Timoshenko beam elements (Kim et al., 2013). The wind turbine structures (i.e., blades, shaft, tower) are represented by a number of bodies, which are defined as an assembly of Timoshenko beam elements (Larsen et al., 2013).The aerodynamic part of the code is based on the blade element momentum (BEM) theory, extended to handle dynamic inflow and dynamic stall (Hansen et al., 2004), among others. In the present study, the HAWC2 turbine model is based on the structural and aerodynamic data of the Siemens SWT2.3-93 turbine and is equipped with the original equipment manufacturer controller.

References:

Hansen, M., Gaunaa, M., and Aagaard Madsen, H.: A Beddoes-Leishman type dynamic stall model in state-space and indicial formulations, 2004

Kim, T., Hansen, A. M., and Branner, K.: Development of an anisotropic beam finite element for composite wind turbine blades in multibodysystem, Renewable Energy, 59, 172–183, https://doi.org/10.1016/j.renene.2013.03.033, 2013.

I classified this as major revisions, because the first issue could change the results or modify the research questions and conclusions, and the other comments point at an incomplete manuscript. Language: I noticed only a few typos or grammar mistakes in the manuscript with the disclaimer that I am not a native English speaker. Parts of the manuscript would benefit from structuring with subsections and paragraphs (sections 1, 3.4, 4.1, and 4.3 specifically).

Specific comments
- Page 1, line 8-9: This sentence reads to me, that lidar-based load predictions in waked conditions are compared against the mast-based predictions from a mast located in the free wind at the same time. But I believe that the intended meaning is, that the uncertainty of lidar-based predictions against on-board sensors in waked conditions are compared with the uncertainty of mast-based predictions against sensor data in free wind conditions.

The following text was replaced in the abstract on page 1 lines (8-9)

From: "The uncertainty and bias of aero-elastic load predictions are quantified against wind turbine on-board sensor data. We consider mast-based load assessments in free wind as a reference case and assess the uncertainty in lidar-based power and load predictions when the turbine is operating in partial- and full-wake"

To: "We assess the uncertainty of lidar-based load predictions against wind turbine on-board sensors in wake conditions and compare it with the uncertainty of lidar-based load predictions against sensor data in free wind."

- Page 1, line 10: Why is only the smallest increase of the relative error given? What was the largest observed increase?

In order to write a concise and short abstract, we reported the best reachable accuracy of load predictions. However, to show the sensitivity of the error due to inflow wind conditions, we now include the range of the error based on the unfiltered turbulence measures with turbulence length scales as for free wind and waked situations.

The following text is replaced in the the abstract on page 1 lines (10)

From: "Compared to the reference case, the simulations in wake conditions lead to an increase of the relative error as low as 4%"

To: "Compared to the free-wind case, the simulations in wake conditions lead to increased relative errors (4–11%)."

- Page 1, lines 10-11: How do they impact the predictions (e.g. do low wind speed lead to a better uncertainty of the prediction or the opposite?)

The analysis in Fig. 11 shows how uncertainty varies as function of the inflow mean wind speed. The results show that low wind speeds lead to higher bias and uncertainty of the predictions. We also mention it in the text (see Page 20, line 3). This trend is noticeable for power predictions, as the power curve is steeper at low wind speed compared to the curve above rated wind speed (Fig. A.1 in the appendix shows the normalized power curve of the turbine). Further, as the power levels are function of the cube of the wind speed, a small error in the estimated wind speed can lead to a significant bias in the power predictions, particularly at low wind speeds.

The results in Table 2 and 3 show the impact of both turbulence intensity and turbulence length scale on the load predictions. We discuss this impact in "Sect. 4.3 Load simulation results" on page 17 lines 20–32.

We believe that information regarding how the wind field characteristics impact the load predictions and their uncertainty should be found in the related sections in the manuscript. Following the reviewer comment regarding Sect. 4.3, we did an extensive re-structuring of the section, where we clarify the dependency of the uncertainty of predictions on the inflow conditions (see below for more details).

We address the reviewer comment in Sect. 4.4, and we divide the section as into Sect. 4.4 Sensitivity analysis and Sect. 4.4.1 Uncertainty distribution as function of wind speed,

- Fig. 2: The remainder of the manuscript often uses x/D as the upwind distance relative to the turbine with the lidars (opposite to here, where it is the downwind distance from the turbine that causes the wake).

This was corrected. The figure is now replaced by the one below.

- Page 5, eq. (1): Within the wake, the wind speed profile is not following a power law profile.

[Figure]

Therefore, I am not convinced that the shear exponent resulting from a partially or fully waked inflow is meaning full. Since the inflow parameters are later used as input for the simulations, the simulated conditions can be expected to be very different from the conditions experienced by the wind turbine. For the CW lidar it seems possible to retrieve spanwise fields of the longitudinal mean velocity provided with some interpolation. Is it possible to initialize the simulations with them instead? As it is, the approach would be better suited to answer what errors are entailed by applying procedures developed for free stream conditions to waked conditions.

Please, see response to major comment nr. 1.

- Page 6, line 2: The simulations should be introduced in more detail.

Please, see response to major comment nr. 3.

- Page 7, line 10: I assume that the assumption of homogeneity is referring to horizontal homogeneity only and not including vertical homogeneity?

The sentence relates to Eq. 5. The inhomogeneities, as wind shear, are introduced after the statement in Eq. 6.

- Fig. 4: Why is the cup anemometer and not the sonic anemometer used for the turbulence measurements? I would expect that the standard deviation from a sonic anemometer is better since it is not affect by cross-contamination and inertia.

The cup is mounted at 80 m, which is the hub height. The sonic is mounted at 76 m. Previous work on the characterization of wind conditions at the NKE site that included wind speed and turbulence showed a discrepancy between the sonic- and cup-based mean wind speed of 2.6% and about 12.3% regarding the longitudinal velocity variance (Peña et al., 2017). To reduce the uncertainty of the mast-based and lidar-based wind characteristics, we choose the cup anemometer for this analysis. The same approach was used in Dimitrov et al. (2019).

The following text is added on page 10, line 8.

Previous work on the characterization of wind conditions at the NKE site that included wind speed and turbulence showed a discrepancy between the 76 m sonic- and 80 m cup-based mean wind speed of 2.6% and about 12.3% regarding the longitudinal velocity variance (Peña et al., 2017). To reduce the uncertainty of the mast-based and lidar-based wind characteristics, we choose the cup anemometer at 80 m, which is the hub height, for this analysis.

- Eq. (14): Why are only the positions B1 and B2 in the upper half of the rotor considered and not the beams B3 and B4 in the lower half?

Preliminary work (Peña et al., 2017) showed that the lidar availability highly reduces when using the bottom beams. This was simply because the lenses of those bottom beams (B3 and B4) became dirty due to contamination from the cleaning system of the CW lidar.

The following text is added on page 11, line 3.

Preliminary work (Peña et al., 2017) showed that the lidar availability highly reduces when using the bottom beams. Therefore, we use the top beams of the PL lidar for this particular analysis.

- Page 12, line 9: How was it determined that the mast is wake-free?

Some details were described on page 11 line 19. However, we now add an Appendix to describe the application of the wake detection algorithm to the mast measurements.

Appendix A: Wake detection from mast measurements

The wake detection algorithm (see Sect.3.4) is extended to the mast measurements to classify 10-min periods where the mast is in free or wake situations. For this purpose, turbulence observations from the cup anemometer at 80 m and vertical wind shear computed using the measurements from the cup anemometers at 57.5 and 80 m are used as wake detection parameters. Their $99th$ percentiles are used as conservative thresholds to characterize the limits of the normal range of the site-specific free wind conditions. The resulting thresholds are $TI_{mast,99} = 0.20$ and $\alpha_{mast,99} = -0.02$. If one of the two limits is exceeded within a 10-min period, the mast is considered in wake conditions and shown with green markers in Fig. 1.

- Fig. 6: As I understood from section 3.1, the effects of the induction zone were removed from the lidar measurements in the wind field reconstruction and the text mentions that the left panel shows reconstructed mean wind speeds (page 12, line 12). Therefore, I am wondering why the effects of the induction zone are present in the lidar data or whether the shown data is based on eq. (5) and not eq. (6)?

We corrected and rephrased the text. In the analysis shown in Fig. 6, we investigate how the lidar-based wind characteristics varies as function of the upstream ranges in free, partial- and full-wake situations. For this particular analysis, we disregard induction effects.

The following text is now added on page 12, line 5:

[Figure]

Figure 1: Left and middle: 10-min observations of the turbulence intensity and vertical wind shear at the mast as function of turbine yaw position. Free wind conditions relative to the mast are identified with grey markers, and waked situations with green markers. Right: PL-estimated 10-min wake detection parameter $TI_{LOS,B1}$. Detected wake situations of turbine T04 are shown with coloured markers: wake-free (grey), partial-wake (blue) and full-wake (red). The 10-min periods, where the mast is affected by wakes are shown in green markers.

From: "We observe these effects through the PL and CW lidar-estimated wind speed, turbulence and shear exponent in Fig. 6. Here, the slope of a linear regression model between the free wind mast-measured and lidar-estimated wind parameters in free-, partial- and full-wake situations are shown as function of the upstream distance from the rotor."

To: "Here, the slope (m) of a linear regression model between the free wind mast-measured and lidar-estimated wind parameters in free-, partial- and full-wake situations is shown. In this particular analysis, the lidar-based wind parameters are derived from Eq. (6) evaluated a different upstream distances from the rotor without including induction effects."

- Fig. 6: I am confused, because the axis labels state the ratio between lidar measurements and mast measurements, but the caption states the slope of a linear regression between.

We changed the figure and provided the corrected axis label.

- Page 14, lines 12-14: From the text I understand that for each wind speed bin an ensemble averaged spectrum was computed and to each of those spectra the model is fitted to estimate L (i.e. for each wind speed bin a separate L is computed). However, that does not line up with single spectrum presented in Fig. 7 (left) and three length scales reported in line 24.

This was not correctly described in the manuscript and we replaced the text accordingly. Also, to improve the readability of the manuscript, we divided section 4.1 in: Section 4.1 Wake effects on reconstructed wind parameters and Section 4.2 Wake effects on turbulence spectra properties.

The following text is now replaced in Page 14, lines 12–14:

[Figure]

From: "The 10-min time series of radial velocity are classified into wind speed bins and the spectra are ensemble-averaged over each wind speed bin. Then, the parameter L is fitted to the ensemble-averaged spectrum weighted on number of samples in each bin.

To: "The 10-min time series of radial velocity are classified into free-, partial- and full-wake situations and the spectra are ensemble-averaged over all conditions within each class. Then, the parameter L is fitted to the ensemble-averaged spectrum."

- Page 14, lines 17-18: Since the shown spectra are normalized with their respective variance, I don't see this from Fig. 7 (left) and only from Fig. 6 (middle).

The measured and theoretical spectra are normalized over their respective variance under each condition (wake-free, partial wake and full wake). Without normalization, the curves will be all over the place as the energy content under each condition is very different.

- Page 14, line 29: At a given wave number, the turbulence kinetic energy depends on the absolute value of energy spectrum and not its slope. Therefore, I am not understanding this sentence. Also, the term rotor sampling frequency could be explained, because I could not find it in quick search and I am not familiar with it.

We have deleted this sentence as it is confusing. The idea behind this sentence was to explain how the wind field spectra influences the loading conditions on the wind turbine.

- Page 15, line 2: How many bins do you have or what is lowest amount of samples in a bin?

This sentence in the manuscript may mislead the reader, as we do not classify inflow parameters nor simulation results in wind speed bins. Therefore, we have deleted this sentence.

The following text is replaced on Page 15, lines 1-2:

From: "We ensure close to 500 10-min samples, distributed nearly equally among wind speed bins in the range 4–14 m/s, for free-, partial- and full-wake scenarios."

To: "We select around 500 10-min samples for each of the free-, partial- and full-wake scenarios, which are distributed within the wind speed range 4-14 m/s."

- Page 15, line 16 to page 16, line 8: The overall validation approach seems sensible to me. However, in my opinion, it is not clearly written down and has to be pieced together from two different places in the manuscript. Separate subsections for the power and the two bending moment might be help to make it easier to digest. In particular, precise definitions of $\tilde{y}$ and $\hat{y}$ were not provided in this manuscript (I looked at the definitions and explanations in Dimitrov et al. (2019) and hope they are also valid here). Also, the generation of two separate bin averaged wind speed ensemble from the previous section should be recalled here. The difference between the mean and the ensemble average with respect to this data set should be explained explicitly. I believe that an extensive rework of section 3.4 is needed, because I misunderstood an essential part of the validation on my first reading and I believe that was not entirely my own fault.

We now rename section 4.3 from "Load simulation results" to "Load validation procedure" and re-arrange Section 4.3 as:
- Section 4.4 Load validation procedure
- Section 4.4.1 Power predictions
- Section 4.4.2 Extreme load predictions
- Section 4.4.3 Fatigue load predictions

We also rephrase section 4.4 to provide a better description of the validation approach as:

[revised manuscript text omitted]

- Page 15, line 18: Check equation. The square should be outside of the bracket.

This is now corrected.

- Page 15, line 19: Check equation. Missing a square.

This is now corrected.

- Page 17, line 8: I am not yet understanding why this bias ratio was chosen as an indicator for the behavior of the uncertainty. With a mast-based reference case, there are two influencing factors with (1) the differences between mast vs. lidar and (2) waked vs. free wind conditions. Why not use lidar-based predictions for free wind conditions as the reference? Then only waked vs. free wind conditions remain, which would make interpretation easier.

Following the suggestion of the reviewer, we changed the reference case, so it is now the lidar-based load predictions in free wind conditions.

Also, in later occurrences of the bias ratio it is not stated whether it is an over- or underestimation (I am assuming that both 1.02 and 0.98 would be given as 2%).

We added the signs to differentiate over- and under-estimation and also specified whether is under- or over-prediction.

- Page 17, line 20: The section is already quite loaded. It might be worth to consider to separate the filtered and unfiltered comparison from the rest of the section. The same might be considered for the length scale.

By re-structuring Section 4.3 into four subsections, as described above, Sect. 4.4.3 Fatigue load predictions will be easier to read.

- Page 18, lines 11: I believe it should be "the increased effect".

This is now corrected.

- Page 18, lines 12: I believe it should be "higher".

The first entry "high" describes the fact that we measure high turbulence levels in the wake; the second entry "higher" work as a comparative, in this case between the turbulence levels under wake- and free conditions.

- Page 18, line 15: Why is the effect of L in the sensitivity analysis minor, but it had an effect of up to 15% in the previous section (Table 2)?

The results in Fig. 10 should be interpreted as the coefficients of a linear regression model between the inputs and the output sensors. The regression models are fitted to a dataset of 850 10-min periods, which are characterized by different inflow conditions (i.e. wind speed, turbulence, shear, etc.) as measured by the PL lidar and the L parameter, which was varied between the defined boundaries (see figure below for the partial-wake). Note that we show the absolute input values in the figure, although the inputs are normalized such that their values are scaled between zero and one, when fitting the regression model. The resulting coefficients give an estimation of the slope of a linear model between inputs and outputs. If a strong dependency is seen between output and inputs, as for the mean wind speed and turbulence in the figure below, a strong sensitivity of output sensor to the input is indicated. In the case of L, we do not see a linear relationship, but rather a scatter plot. This is because, while we vary L, the wind speed and turbulence levels are also varied, and the impact of the latter two is higher to that of L. However, when all inputs are fixed and only L varies, as the results in Table 2 and Table 3, we can estimate the effective influence of L on the predictions.

[Figure]

Figure 2: Scatter plot of the normalized fatigue loads at the blade-root as function of inflow wind conditions (mean wsp, turbulence and turbuelence length scale L)

- Page 22, lines 10-11: Consider rephrasing this sentence, because it can be understood in two ways.

The sentence "The effective turbulence approach under predicted fatigue load levels at spacings larger than 5 D (Schmidt et al., 2011)." is now deleted, in case this is what the reviewer was pointing at

- Page 22, lines 27-28: Reiterating a previous comment, I am not convinced that the power law should be applied within the wake, because the wind profile has a different shape. The conclusion regarding the vertical wind profile here is based on the sensitivity to the shear exponent and I am not convinced that it holds for the same reason, because the simulated wind fields might have been

very different from the real wind field. Also, I am confused regarding the horizontal wind profile, because I cannot remember that horizontal gradients were accounted for in the wind field retrieval. Is that referring to the wind veer maybe?

Please, see the response to major comment nr. 1. We deleted the sentence regarding the horizontal gradient and now discuss the influence of turbulence intensity and turbulence length scale on the power productions levels, as shown below.

The following text is replaced on Page 22, lines 27-28:

From: "Power production levels under wake conditions were strongly related to the mean wind speed at hub height, whereas the vertical and horizontal wind profiles had negligible effects on those levels"

To: "Power production levels under wake conditions were strongly driven by the reconstructed wind speed at hub height, whereas turbulence intensity as well as turbulence length scales had negligible effects on those levels."

- Page 29, lines 23-24: The reference of Dimitrov et al. (2019) seems to be out of place and should appear after Dimitrov et al. (2018) assuming the sorting hierarchy is first author alphabetical followed by year.

This is now corrected.

---

## Author Response (AR2)

**Author response to reviewer 1**

The authors response is shown in red
Changes implemented in the new version of the manuscript are in blue

We thank the reviewer for the additional comments, which we consider important to further improve the manuscript. Here our response to each of them.

- Original reviewer comment: "More motivation for the presented work should be included. For example, I was wondering how accurate current IEC-recommended methods for estimating loads in waked conditions are, and if there is a clear need to improve these methods using lidar-measured wind conditions. Additionally, I was wondering why the authors did not compare the accuracy of the load predictions to the accuracy when using these IEC standard wake modeling methods. It was not until the end of the paper (last paragraph of Section 5) that these topics were discussed. I believe this material should be moved to the introduction to better motivate the research and to explain the scope of the current work (i.e., that comparison with DWM, etc. is not part of the current work)."

Author response: "Following the comment of the reviewer, we now provide further motivation on the need to improve load validation approaches in wakes, and clarify the scope of the work along the manuscript. Further, "Sect. 2.1 Requirements for load validation in wakes" is fully dedicated to define the scope of the current work and introduce the reader with the elements of the load analysis.

The following text is added on page 2 line 9: ..."

Moving this discussion to the introduction has strengthened the paper, but it is still repeated in the discussion. To avoid duplicating material, perhaps just refer back to the intro in the discussion section?

Following the suggestion of the reviewer, we delete the repeated paragraph in the discussion and refer to the Introduction.

The following text is now added: "...These models ensure an acceptable level of accuracy, robustness and computational cost. Previous studies carried out load validation using the effective turbulence model and the DWM model, which are recommended in the IEC 61400-1. As described in Sect. 1, results from these studies show deviations of the same or even larger order of magnitude..."

- Original reviewer comment: "Many of the paragraphs throughout the manuscript are very long, often an entire page. Organizing them into smaller paragraphs would greatly improve the readability. For example, pg. 2, pg. 11, pg. 14, pg. 17."

Author response: "To improve readability, we restructured the manuscript as following:..."

The changes by the author help improve the organization and readability of the paper, but there are still many paragraphs that are too long and should be divided into smaller pieces for easier reading. For example, Section 3.4 is one very long paragraph. Same with Section 4.2. The first paragraph of Section 5 should also be split into smaller parts.

We have now divided the large paragraphs in smaller parts.

- Original reviewer comment: "4) Table 1: Can you explain the difference between uhub (mean wind speed at hub height) and U (in the TI definition)? Or are these the same and could be written with the same symbol?"

Author response: "We now replace the U with u in Table 1 and throughout the manuscript."

This is a good change, but it makes it unclear if $\bar{u}_{hub}$ and $\bar{u}$ are the same quantity, or if $\bar{u}$ varies with height. TI is written as $\sigma_u/\bar{u}$ (instead of $\sigma_u/\bar{u}_{hub}$), which makes it seem like TI varies with height in your wind field model. Can you clarify in the text if this is the case?

The TI used as input for the load simulations is computed as $TI = \sigma_u/\bar{u}_{hub}$. Therefore, we now correct the convention in Table 1.

- Original reviewer comment: "8) Pg. 7, ln. 17: "The parameters (uhub,: : :" Because of the induction zone model, I imagine the induction factor is also a parameter that is estimated. Is this true?"

Author response: "This is correct, the Cind is now included in the sentence."

I think it makes more sense to list the induction factor $a_0$ as the parameter instead of the function $C_{ind}$.

We have now replaced $C_{ind}$ with $a_0$ in the text.

- Original reviewer comment: "15) Pg. 10, ln. 8: "The filtered turbulence derived from CW and PL lidars are plotted: : :" Which beams are used to derive the turbulence?"

Author response: "To derive the filtered turbulence estimates from both the CW and PL lidars, we used all the beams and ranges. To derive the unfiltered turbulence from the CW lidar, we use only measurements at 1.3 D.

We replaced the text in the manuscript on page 10 line 8:

From: "The unfiltered turbulence derived from the CW and PL lidars are plotted respectively in Fig. 4 (left and middle), whereas the unfiltered turbulence derived from the CW lidar is shown in Fig. 4 (right)."

To: "The unfiltered turbulence derived from CW and PL lidars, using all the ranges and beams, are plotted respectively in Fig. 4 (left and middle), whereas the unfiltered turbulence derived from the CW lidar measurements at 1.3 D are shown in Fig. 4 (right).""

Thanks for clarifying, but please also explain in the text how the multiple measurement points are combined to estimate TI. For example, are the TI values derived from each measurement point averaged together to form a single TI estimate?

We derive an estimate of $\sigma_u^2$ by combining multiple lidar measurements according to the 'first approach' described in Sect. 3.3. The approach is based on the Eqs. 6, 12, 13 and 14, and it is described in the text (page 10 line 20-23). "It follows that the ratio in Eq. 14 identifies the relation between the LOS variance and the wind field variance in the longitudinal direction. Eventually, the variance of the wind field is computed by scaling the variance of the LOS residuals with the reciprocal of the filtering ratio estimated using Eq. 14. The procedure is described in details in Dimitrov et al. 2019." For example:

The LOS residuals $(u'_{los})$ are computed as the difference between the lidar measurements $u_{los}$ and the mean LOS field evaluated through the wind model defined in Eq. 6 that we denote as $\bar{u}_{los}(\bar{u}_{hub}, \alpha, \Delta\varphi, \bar{\varphi}, a_0)$:

$$u'_{los} = u_{los} - \bar{u}_{los}(\bar{u}_{hub}, \alpha, \Delta\varphi, \bar{\varphi}, a_0) \tag{1}$$

If we scale the variance of the LOS residuals $\text{var}(u'_{los})$ from Eq. 1 with the reciprocal of the ratio in Eq. 14, we obtain $\sigma_u^2$ as:

$$\sigma_u^2 = var(u') = var(u'_{los}) \cdot \left( r(Z_r, L, \Gamma, \psi_y, \psi_z)^2 \left( \boldsymbol{T_{LOS}} \boldsymbol{C} \boldsymbol{T_1} \frac{\boldsymbol{R}}{\sigma_u^2} \boldsymbol{T_1^T} \boldsymbol{C^T} \boldsymbol{T_{LOS}^T} \right) \right)^{-1} \tag{2}$$

This is the approach used to derive the filtered turbulence estimates by combining multiple lidar measurements as proposed in Dimitrov et al. 2019. For clarification, we do not derive TI for each measurement point and average them, but we derive a single $\sigma_u^2$ estimation from Eq. 2, which combines multiple measurements at once.

In order to clarify the methodology, we replace the following text in Sect 3.3. "It follows that the ratio in Eq. 14 identifies the relation between the LOS variance and the wind field variance in the longitudinal direction. As described in Dimitrov et al. 2019, the LOS residuals $u'_{LOS}$ are calculated as the difference between the LOS measurements $u_{LOS}$ and the mean LOS field $\bar{u}_{LOS}(\bar{u}_{hub}, \alpha, \Delta\varphi, \bar{\varphi}, a_0)$ obtained from Eq. 6 as:

$$u'_{LOS} = u_{LOS} - \bar{u}_{LOS}(\bar{u}_{hub}, \alpha, \Delta\varphi, \bar{\varphi}, a_0). \tag{3}$$

Eventually, $\sigma_u^2$ is derived by scaling the variance of the LOS residuals from Eq. (3) with the reciprocal of the filtering ratio estimated using Eq. 14. As the filtering ratio is evaluated for each LOS direction, we can combine multiple lidar measurements to estimate $\sigma_u^2$. The procedure is described in details in Dimitrov et al. 2019."

-Pg. 10, ln. 23: "The second approach avoids..." Since you are switching topics and talking about the 2nd approach after a few paragraphs focusing on the 1st approach, it makes sense to start a new paragraph here.

This is done as suggested.

- Pg. 16, ln. 12: "When compared to the filtered turbulence, the unfiltered estimations show a significant reduction by 6%..." In Figs. 8 and 9 when comparing the filtered and unfiltered turbulence estimates, it appears there might be an error in the plots. The filtered and unfiltered cases should be for CW measurements, but in the plots the differences between blue and black are along the pulsed lidar ("y") axis, while the corresponding values along the CW axis remain the

same. I think the opposite should be true. Is this an error?

There was an error in the plot. The blue dots were actually referring to CW unfiltered (y-axis) and CW filtered (x-axis). We have corrected the plots by showing the comparison between the PL filtered (y-axis) and CW unfiltered (x-axis).

- Original reviewer comment: "21) Pg. 14, ln. 32: "It can be noticed that broadening effects are present only in b3: : :" Where is the evidence of broadening effects? Wouldn't this require the velocity spectrum in freestream conditions for comparison?"

Author response: "We added the Doppler spectrum for a wake-free case with similar inflow conditions."

This is a nice addition to the plots. But in the text, please describe in what sense the inflow conditions are similar for the measurement periods of the different Doppler spectra.

The following text is added: "For this comparison, we select three 10-min periods with similar inflow conditions measured at the mast, thus $\bar{u}_{hub} \sim 9$ m/s and $\sigma_u/\bar{u}_{hub} \sim 0.11$."

- Original reviewer comment: "22) Pg. 15, ln. 18: Please compare with the coefficient of determination equation in Dimitrov et al. 2019. It appears there are some typos in the equation listed here.

Author response: "The coefficient of determination is now corrected"

There is still an error in the coefficient of determination equation. Both expectations should be the expected value of the measured statistics (y hat).

The coefficient of determination is now corrected.

- Original reviewer comment: "23) Pg. 16, ln. 5: Why is only MXBCmin investigated, as opposed to MXBCmax?"

Author response: "This is due to the convention used in the strain gauges. The increasing flapwise bending moment results in negative loading. We keep the same convention as in (Dimitrov et al., 2019)."

It would be good to provide this explanation in the text and also explain what direction negative loading is in and why this is the direction of the maximum loading.

The following text is now added: "Note that given the strain gauges convention, the increasing flapwise bending moment results in negative loading, thus we refer to MxBCmin as the extreme loads (Dimitrov et al., 2019)."

- Original reviewer comment: "27) Pg. 18, ln. 7: "The obtained linear regression coefficients for Power: : :" This is a nice analysis. Are the trends the same for MXBCmin and MXTBDEL (which aren't shown)?"

Author response: "We show the linear regression coefficients for all the channels in the figure below."

It would be good to briefly mention in the text if the trends for the coefficients for for MxBC min and MxTB DEL are the same or different as the other channels since they are not shown in the plots.

The following text is added "Similar trends are obtained for $M_{xBC_{min}}$ and $M_{xTB_{DEL}}$ (not shown)."

- Pg. 23, ln. 17: "Although we demonstrate a low sensitivity of the loads to the shear exponent..." It is hard to understand the meaning of this sentence. Are you trying to say that "horizontal shear" or a "horizontal velocity gradient" is envisioned to more appropriately account for wake-affected velocity gradient profiles...?

Based on previous comments, we divided the text in smaller paragraphs, which helps the reading. Thus, a paragraph focusing on the wind modeling assumption is now created in the discussion. Further, we add more details in the text to clarify the sentence.

The following text is added: "Although we demonstrate a low sensitivity of the loads to the shear exponent for all the analysed sensors (see Fig. 10), it is envisioned to more appropriately account for wake-affected velocity gradient profiles, which include a wake shape function and the contribution of the meandering, and determine whether or not this will significantly improve the accuracy of power and load predictions."

Pg. 25, ln. 9: "...load predictions of the order of 4% compared to the free-wind case" Kind of confusing. Do you mean 4% "higher" compared to the free-wind case?

The following text is replaced: "Furthermore, simulations with low turbulence length scales led to an underestimation of blade-root fatigue load predictions by 4% compared to on-board sensors, while free wind situations were unbiased."

**Author response to reviewer 2**

The authors response is shown in red

We thank the reviewer for the technical comments provided, which we consider important to further improve the manuscript.

Technical comments:

- Section 4.2 and 5 could be improved by paragraphs breaks.

Following the reviewer suggestion, we divide the text in smaller paragraphs.

- Page 25, line 1: "compared to the free wind conditions" could be misunderstood and something along the lines of "compared to on-board sensors, while free wind conditions were unbiased" might be clearer.

*The text has been corrected following the suggestion of the reviewer.*

- Figure A.1: The coordinate system indicated is rotated by 180 around the z-axis compared to the other figures of the manuscript.

*This is correct, we now rotate the coordinate system.*

[revised manuscript text omitted]